

# High-resolution airborne imaging DOAS-measurements of $NO_2$ above Bucharest during AROMAT

Andreas Carlos Meier[1], Anja Schönhardt[1], Tim Bösch[1], Andreas Richter[1], André Seyler[1], Thomas Ruhtz[2], Daniel-Eduard Constantin[3], Reza Shaiganfar[4], Thomas Wagner[4], Alexis Merlaud[5], Michel Van Roozendael[5], Livio Belegante[6], Doina Nicolae[6], Lucian Georgescu[3], and John Philip Burrows[1]

[1]Institute of Environmental Physics, University of Bremen, Otto-Hahn-Allee 1, 28359 Bremen, Germany
[2]Institute for Space Sciences, Free University of Berlin, Carl-Heinrich-Becker-Weg 6-10, 12165 Berlin, Germany
[3]"Dunarea de Jos" (University of Galati), Str. Domneasca 111, Galati 800008, Romania
[4]Max-Planck-Institute for Chemistry, Hahn-Meitner-Weg 1, 55128 Mainz, Germany
[5]Royal Belgian Institute for Space Aeronomie (BIRA-IASB), Avenue Circulaire 3, 1180 Brussels, Belgium
[6]National Institute of R&D for Optoelectronics (INOE), Magurele, Street Atomistilor 409, Magurele 77125, Romania

*Correspondence to:* A. C. Meier (ameier@iup.physik.uni-bremen.de)

**Abstract.** In this study we report on airborne imaging DOAS measurements of $NO_2$ from two flights performed in Bucharest during the AROMAT campaign (Airborne ROmanian Meeasurements of Aerosols an Trace gases) in September 2014. These measurements were performed with the Airborne imaging Differential Optical Absorption Spectroscopy (DOAS) instrument for Measurements of Atmospheric Pollution (AirMAP) and provide nearly gapless maps of column densities of $NO_2$ below

the aircraft with a high spatial resolution of better than $100\,\mathrm{m}$. The airmass factors, which are needed to convert the measured differential Slant Column Densities (dSCDs) to Vertical Column Densities (VCDs) have a strong dependence on the surface reflectance, which has to be accounted for in the retrieval. This is especially important for measurements above urban areas, where the surface properties vary strongly. As the instrument is not radiometrically calibrated, we have developed a method to derive the surface reflectance from measured intensities at the aircraft. This method is based on radiative transfer calculation

with SCIATRAN and a reference area for which the surface reflectance is known. While surface properties are clearly seen in the $NO_2$ dSCD results, this effect is successfully corrected for in the VCD results. Furthermore we investigate the influence of aerosols on the retrieval for a variety of aerosol profiles that were measured in the context of the AROMAT campaigns. The results of two research flights are presented which reveal distinct horizontal distribution patterns and strong spatial gradients of $NO_2$ across the city. Pollution levels range from background values in the outskirts located upwind of the city to about

$4 \times 10^{16}\,\mathrm{molec\,cm^{-2}}$ in the polluted city center. Validation against two co-located mobile car-DOAS measurements yields good agreement between the datasets with correlation cofficients of R=0.94 and R=0.85, respectively. Estimations on the $NO_x$ emission rate of Bucharest for the two flights yield emission rates of $15.1 \pm 9.4\,\mathrm{mol\,s^{-1}}$ and $13.6 \pm 8.4\,\mathrm{mol\,s^{-1}}$, respectively.



# 1 Introduction

$NO_x$, the sum of NO and $NO_2$, plays a key role in the chemistry of the atmosphere. In the troposphere, it is produced by natural and anthropogenic precesses, such as fossil fuel combustion, biomass burning, lightning and bacterial degradation of fertilizers. Because its main source are combustion processes, $NO_x$ can serve as an indicator of anthropogenic pollution. The release of $NO_x$ by anthropogenic activity leads to adverse effects on the environment, such as the formation of tropospheric ozone, eutrophication and acid rain, as well as negative impacts on human health. Consequently there is a large societal interest in knowing the amounts and spatial distributions, sources and sinks of $NO_x$.

$NO_2$ exhibits characteristic absorption structures in the UV/visible spectral range, enabling the application of the DOAS (Differential Optical Absorption Spectroscopy) method to measure column densities of this trace gas (Platt and Stutz, 2008). Using this DOAS technique, tropospheric $NO_2$ can be measured from space-borne satellite instruments, such as GOME (Burrows et al., 1999), SCIAMACHY (Burrows et al., 1995; Bovensmann et al., 1999), OMI (Levelt et al., 2006) or GOME-2 (Munro et al., 2016). While the satellite measurements demonstrate their value in providing a global picture of the $NO_2$ distribution, their spatial resolution of several tens of kilometers is too coarse to investigate the horizontal $NO_2$ distribution on smaller scales such as individual cities.

Measurements of $NO_2$ on smaller scales are usually achieved by ground-based instruments of different types. Stationary measurements provide long time series of data and facilitate the investigation of diurnal and seasonal variability as well as long-term trends. They can thus also be used for validation of satellite data and chemical models (Eskes et al., 2015). Mobile DOAS measurements are also performed, mostly from cars, offering the advantage of covering a large area at comparably low costs (Wagner et al., 2010; Ibrahim et al., 2010; Constantin et al., 2013; Shaiganfar et al., 2015). However, practical issues such as having to follow the pattern of roads or traffic jams limit the spatial extent of these measurements.

Application of DOAS instruments on airborne platforms may bridge the gap between ground-based and satellite measurements, because they can cover a large area in a relatively short time with less limitations on the covered area and measurement pattern. Airborne DOAS measurements were, for example, performed using the AMAX-DOAS (Airborne Multi AXis DOAS) technique facilitating the validation of satellite instruments and the retrieval of trace gas profiles (Wang et al., 2005; Heue et al., 2005; Oetjen et al., 2013; Baidar et al., 2013) or the measurement of shipping emissions (Berg et al., 2012). Small instruments have also been applied on ultralight aircrafts (Merlaud et al., 2012).

In more recent years, imaging DOAS (iDOAS) instruments were developed. Lohberger et al. (2004) demonstrated the applicability for trace gas retrievals in a ground-based setup. Installed on aircrafts, these systems enable the creation of maps of the horizontal trace gas distribution. iDOAS measurements of anthropogenic point source emissions were performed by Heue et al. (2008) and by Schönhardt et al. (2015) above the South African Highveld plateau and above a German power plant, respectively. General et al. (2014) performed measurements of volcanic emissions at Mt. Etna, Italy. Urban $NO_2$ distributions were measured by Popp et al. (2012) above the city of Zürich, Switzerland, and by Lawrence et al. (2015) above the city of Leicester, England.





Measurements of $NO_2$ in urban areas are essential to understand the distribution of pollutants and to develop strategies for the mitigation of air pollution events. Thus, many cities have installed in-situ systems to monitor air quality at ground level. However, these in-situ stations have a sparse spatial sampling. They are often on road sites for good reasons and thus directly impacted and dominated by automobile exhaust plumes. Mapping of $NO_2$ distributions above cities with airborne iDOAS may provide a holistic picture on pollution levels across the city.

In this study we present airborne imaging DOAS measurements of $NO_2$ performed in the framework of the AROMAT campaign (Airborne ROmanian Measurements of Aerosol and Trace gases), which took place in September 2014 in Romania. One purpose of this campaign was to test and compare state-of-the-art instruments in preparation for the validation of the upcoming Sentinel-5 precursor satellite mission (S5p, (Veefkind et al., 2012)). The campaign comprised a variety of remote sensing and in-situ instruments used to measure atmospheric composition and aerosol load, facilitating a detailed characterization and comparison of the different measurements. The campaign had two target sites, the city of Bucharest and a power plant in Turceni, the latter representing an isolated point source in a rural area. However, this work concentrates on measurements in the urban area of Bucharest.

The study focuses on the retrieval of accurate VCDs (Vertical Column Densities) by applying scene specific surface reflectances determined from intensities measured by the same instrument, which is not radiometrically calibrated. The retrieval and application of surface reflectances i.e. the method to retrieve the VCD is explained. The sensitivity of the VCD retrieval is investigated for several aerosol load scenarios which are based on measurements close to Bucharest. Results from two research flights above Bucharest are presented and the dataset is compared to ground-based car-DOAS measurements for validation. In the last section, the emission flux of $NO_x$ is estimated by applying Gauss's divergence theorem to our data.

## 2 The Campaign

### 2.1 The target area

The AROMAT campaign took place in Bucharest in September 2014. Bucharest, located at 44.4°N, 26.1°E, is the largest city and capital of Romania. It has around 1.9 million inhabitants (Central Intelligence Agency, 2016) and covers an area of $228\,\mathrm{km}^2$. According to Constantin et al. (2012), traffic is the dominant source of $NO_x$ in the area, but industrial sources also contribute to $NO_x$ pollution. Nine industrial pollution sources in the area are listed in the European Pollution Release and Transfer Register (E-PRTR), (European Environment Agency), which may have a strong influence on urban pollution levels.

### 2.2 The research flights

Two flights above Bucharest with almost the same flight pattern were performed during the AROMAT campaign, cf. Fig. 1. The flights aimed at producing maps of the $NO_2$-field above Bucharest. To achieve that, parallel tracks were flown with a distance of $1900\,\mathrm{m}$. This pattern provides sufficient overlap of the swaths between adjacent flight tracks to produce a gap-less map. Both flights were performed on a weekday, Monday and Tuesday, respectively. During the research flights, an area of about





## Overview measurements

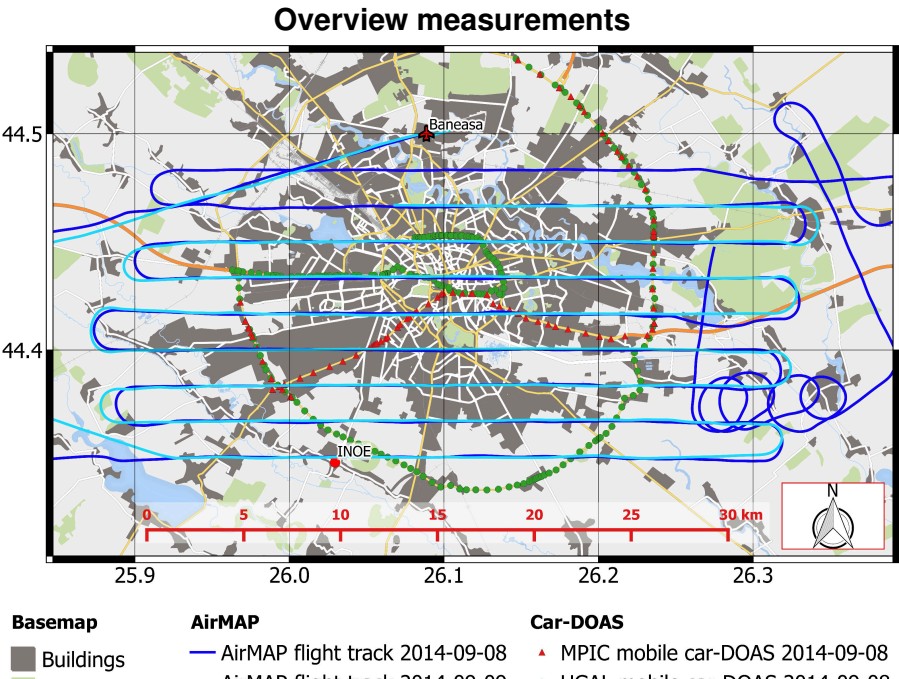

| Basemap | AirMAP | Car-DOAS |
|---|---|---|
| ■ Buildings | — AirMAP flight track 2014-09-08 | ▲ MPIC mobile car-DOAS 2014-09-08 |
| ■ Forest & park | — AirMAP flight track 2014-09-09 | ● UGAL mobile car-DOAS 2014-09-08 |
| ■ Water | | |

**Figure 1.** Map of Bucharest with an overview of the measurements. Blue lines show the flight tracks performed by AirMAP. Circles and triangles mark the measurement locations of the supporting ground-based mobile DOAS systems. No car-DOAS measurements are available on 2014-09-09 because on this day these systems were transferred to the other campaign site Turceni. Additionally shown are the locations of the INOE research institute, where ground based aerosol measurements were performed, and Baneasa Airport.

$560 \, \text{km}^2$ was covered in approximately 1.5 h. The flight on 2014-09-08 started the pattern in the North, whereas the flight on the next day began the pattern in the South. Both flights were performed under cloud free and sunny conditions with low wind speeds ($< 1.5 \, \text{ms}^{-1}$). Further details about the flights are shown in Table 1.

**Table 1.** Overview of the analyzed flights

| | Total | At measurement altitude (3.4 km) | | | |
|---|---|---|---|---|---|
| Day | Flight time [h] | Flight time [h] | Duration [h] | $SZA_{min}[°]$ | $SZA_{max}$ [°] |
| 2014-09-08 | 8.80 – 11.11 | 9.09 – 10.74 | 1.65 | 38.73 | 41.41 |
| 2014-09-09 | 7.42 – 9.20 | 7.89 – 9.20 | 1.31 | 41.3 | 49.83 |

Times are provided in UTC; Local time (EEST) is UTC+3



## 3 Instrument and data acquisition

### 3.1 The AirMAP instrument

The Airborne imaging Differential Optical Absorption Spectroscopy (DOAS) instrument for Measurements of Atmospheric Pollution (AirMAP) has been developed for the purpose of trace gas measurements and pollution mapping. A detailed de-

scription of the instrumental setup, its performance, the viewing geometry and the georeferencing is given in Schönhardt et al. (2015). Thus, the instrument is only briefly described here.

AirMAP is a push-broom UV/vis imager with a wide field-of-view of around 51.7° across track in its current setup, leading to a swath width of about the same size as the flight altitude. Scattered sunlight from below the aircraft is collected with a wide field of view objective. The light is coupled into an imaging grating spectrograph via a sorted fiber bundle, retaining the

spatial information. The fiber bundle consists of 35 individual fibers, that are stacked vertically at the spectrometer entrance slit and are oriented across flight direction in the focal plane of the objective. The dispersed light is imaged onto a FT-CCD (Frame Transfer Charge Coupled Device), Princeton Instruments PhotonMAX 512B. The frame transfer technique of the CCD provides a fast frame rate, because the charges are quickly shifted into a masked storage area for read-out. This procedure allows gap-less measurements, because the next image can be recorded within milliseconds. For data safety reasons, the CCD

readout is interrupted and restarted every few minutes resulting in small measurement gaps. The spectrometer is an Acton 300i imaging spectrograph with a focal length of 300 mm, and an f-number of f/3.9, temperature stabilized at 35°C. The wavelength region can be chosen according to the chemical species of interest, with a spectral coverage of either 41 nm or 86 nm, using a 600 g/mm grating blazed at 500 nm or a 300 g/mm grating blazed at 300 nm, respectively. For the flights above Bucharest described in this paper, the 600 g/mm grating was used for measurements in the visible spectral range (420-461 nm). Table 2

lists important properties of the AirMAP instrument as an overview.

The across-track spatial resolution depends on the flight altitude, whereas the along-track resolution depends of the ground-speed of the aircraft and the exposure time. During AROMAT, AirMAP was installed on a Cessna 207 Turbo aircraft, operated by the Free University of Berlin. The AirMAP instrument as well as the aircraft are equipped with an Attitude and Heading Reference System (AHRS) and GPS sensor, allowing an accurate georeferencing. For typical conditions during the AROMAT

campaign, (flight altitude 3.4 km, ground speed 60 ms$^{-1}$, exposure time 0.5 s), the footprint of one ground pixel is 94×30 m$^2$. With the AirMAP setup it is thus possible to examine the sub-pixel variability within one OMI pixel (13×24 km$^2$ at nadir), or even an individual S5p-satellite pixel (3.5×7 km$^2$ at nadir).

### 3.2 Data preparation and spectral analysis

During the flights, spectra of scattered sunlight from below the aircraft are recorded. The datasets are series of images from

the square CCD-chip with the spectral information on the horizontal axis and spatial information on the vertical axis. In the post-processing, adjacent rows of the CCD are averaged according to the illumination by the individual light fibers. This results in time series of individual spectra for each viewing direction. The spectra are georeferenced according to the Cessna's AHRS data, interpolated to 8 Hz, using a nearest neighbor synchronization of the GPS timestamps. The GPS-altitude is provided



**Table 2.** Properties of the AirMAP instrument during AROMAT

| Parameter | Value |
|---|---|
| CCD dimensions | $8.6 \times 8.6 \, \text{mm}^2$ |
| # CCD pixels | $512 \times 512$ |
| Exposure time | $0.5 \, \text{s}$ |
| Spectrometer type | Czerny-Turner |
| f-number | f/3.9 |
| Focal length | $300 \, \text{mm}$ |
| Grating | $600 \, \text{g} \, \text{mm}^{-1}$ |
| Spectral range | 420-461 nm |
| Spectral resolution (FWHM) | 0.4-1.1 nm |
| $\text{FOV}_{\text{along}}$ | $1.2°$ |
| $\text{FOV}_{\text{across}}$ | $51.7°$ |
| # viewing directions | 35 |

as the altitude above the WGS84 reference ellipsoid and is corrected for altitude above ground level with a digital elevation model (DEM) (European Environment Agency, 2013). Calibration and dark measurements are performed on ground. Spectral calibration is performed using emission lines of a HgCd-spectrum and a high resolution solar atlas (Kurucz et al., 1984). Subsequently, the DOAS method (Platt and Stutz, 2008) is applied to the calibrated spectra. Using an extraterrestrial solar

spectrum as background spectrum in the DOAS analysis, as it is the case for satellite measurements, yields slant column densities (SCDs), which are the number densities of an absorber, integrated along the light path. The background spectrum $I_0$ used in the DOAS analysis of the AirMAP spectra is a 60 s averaged spectrum taken from a scene of the same flight with low absorber abundances. This spectrum may contain small amounts of the absorber. Thus, the results of the DOAS analysis are differential slant column densities (dSCDs), representing the difference of the absorber number densities between the scene

studied and the background spectrum. Because the background spectrum is taken at a clean location, the dSCD is close to the SCD. Because of aberrations in the imaging system, the spectral resolution of the measurements varies across track. Therefore, each individual viewing direction has its own spectral calibration and background measurement. The most important settings used in the DOAS $NO_2$ retrieval are displayed in Table 3.

### 3.3 Gridding of data

To generate composite maps from overlapping measurements of $NO_2$, as well as for the comparison of multiple overpasses, it is necessary to grid the data. To produce these gridded datasets, a simple gridding algorithm was used. If not stated otherwise, a



**Table 3.** Fitted absorption cross-sections and important settings used in the retrieval of $NO_2$ dSCDs

| Molecule / Parameter | Temperature | Reference / Property |
|---|---|---|
| $O_3$ | 223 K | Serdyuchenko et al. (2014) |
| $NO_2$ | 294 K | Vandaele et al. (1998) |
| $O_4$ | 293 K | Thalman and Volkamer (2013) |
| $H_2O$ | 293 K | Rothman et al. (2013) |
| Ring effect | - | Rozanov et al. (2014) |
| Intensity offset | | constant |
| Fit window | | 425-450 nm |
| Polynomial | | quadratic |
| $I_0$ | | 60 s average, per viewing direction |

regular grid with a spatial resolution of $0.0008° \times 0.0008°$, corresponding to $89 \times 64\,m^2$ was defined, which is the approximate size of two subsequent measurement footprints during a level flight, i.e. when the aircraft's pitch and roll angle is small. Measurements, whose pixel center falls into a grid cell are assigned to that grid cell. Multiple measurements in one grid cell are averaged by the unweighted arithmetic mean. This approach was chosen to optimize computation time but may introduce

small biases in the geo-location. Furthermore, when the size of the footprint becomes larger, gaps are introduced between the neighboring grid cells. This effect can be observed in turns when the projected footprint becomes larger.

### 3.4 Differential slant column densities

Figures 2 and 3 show the retrieved differential slant column densities measured during two flights performed on subsequent days, 2014-09-08 and 2014-09-09 along with major roads. The data is gridded to having a spatial resolution of $0.0008°$ x

$0.0008°$. Different light path lengths, caused by different Viewing Zenith Angles (VZA) were geometrically corrected by multiplication with the factor $\cos(VZA)$. The dashed white box shows the region, where the background spectrum, $I_0$, was taken. Slightly negative values occur in the background region as a result of instrumental noise, because the dSCD-values are scattered around zero. In the most polluted areas, dSCD-values of up to $6.1 \times 10^{16}\,molec\,cm^{-2}$ are observed.

The dSCDs show a plume of $NO_2$ spreading south-westwards from the city center. The $NO_2$ field inside the plume shows

small-scale structures with high values. Some of these structures, e.g. the pronounced values at the ring road in the South-West, are not associated with $NO_2$ emissions from traffic, but are related to bright surfaces. Section 4.2 explains the origin of these spatial patterns above bright surfaces and describes the approach used to account for these radiative transfer effects. The results of this correction will be shown in Sec. 6.





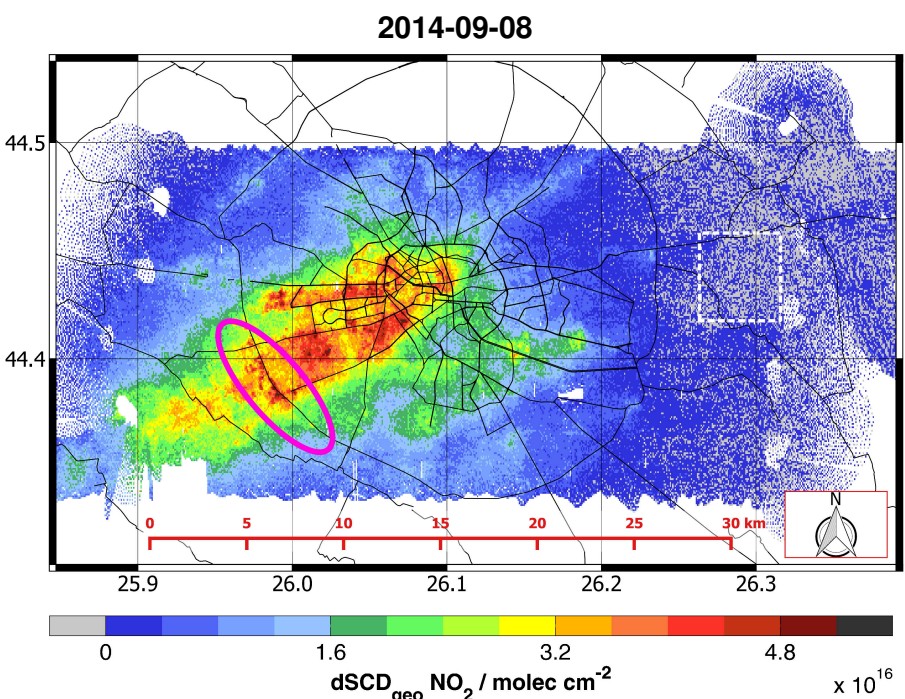

**Figure 2.** Geometrically corrected differential slant column densities of $NO_2$ above Bucharest measured on Monday, 2014-09-08. The white dashed box shows the area, where the background spectrum was taken. Major roads (black lines) are overlaid for orientation. The pink ellipse highlights some of the small-scale structures described in the text.





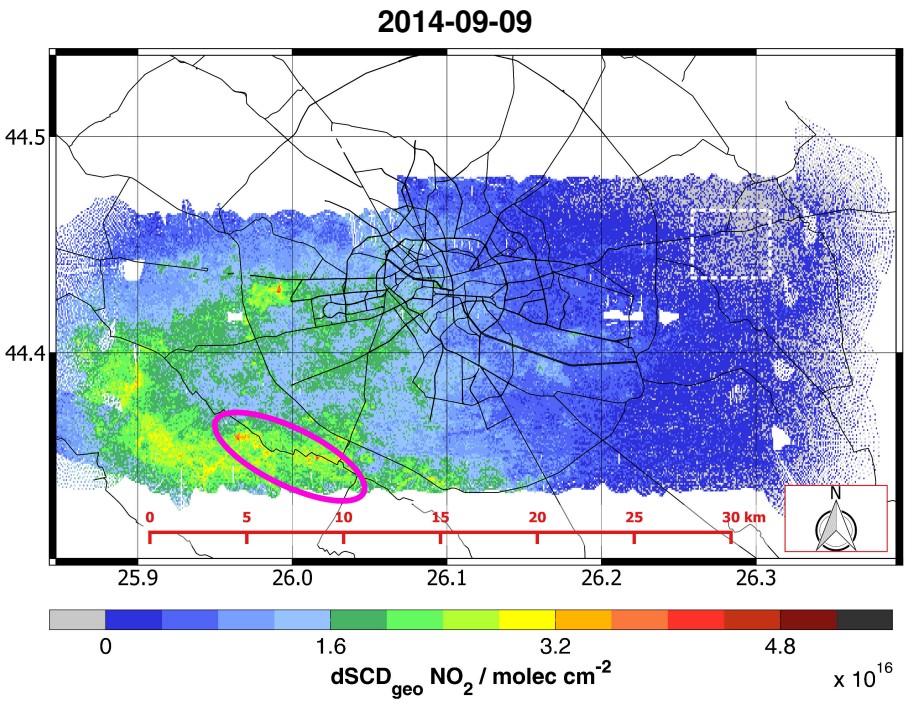

**Figure 3.** Geometrically corrected differential slant column densities of $NO_2$ above Bucharest measured on Tuesday, 2014-09-09. The white dashed box shows the area, where the background spectrum was taken. Major roads (black lines) are overlaid for orientation. The pink ellipse highlights some of the small-scale structures described in the text.





## 4 Derivation of vertical column densities

Using the dSCDs resulting from the DOAS fits, vertical column densities (VCDs), defined as the absorber concentration integrated along the vertical direction, are computed. The conversion of the retrieved dSCD to a VCD enables comparisons of measured trace gas column densities irrespective of the instrument viewing geometry, solar position and surface properties. The alteration of the light path, as compared to a normal to the surface, is usually expressed in the form of an air mass factor (AMF), which is defined as the ratio of slant and vertical column densities:

$$AMF = \frac{SCD}{VCD} \tag{1}$$

### 4.1 Conversion of retrieved dSCD to tropospheric VCD

Satellite platforms have the advantage of being able to measure a solar spectrum without atmospheric absorption as a background spectrum for the DOAS analysis. This is not the case for platforms operating within the Earth's atmosphere. In addition, AirMAP has no option to point into the zenith direction. Thus, the background spectrum for the DOAS retrieval is taken above a rural scene with small $NO_2$ concentrations on the same flight. The following equations describe the conversion of the retrieved dSCDs to tropospheric VCDs ($VCD^{trop}$). The $dSCD$ and the $AMF$ are the trace gas result and the AMF of each individual observation. The superscripts $trop$ and $strat$ refer to tropospheric and stratospheric parameters, respectively. The subscript $0$ refers to conditions of the DOAS background spectrum measurement.

First, the dSCDs are converted to tropospheric slant column densities by correction of the absorber amount in the rural background scene, $SCD_0^{trop}$, and by changes in the stratospheric slant column, relative to the background spectrum, $\Delta SCD^{strat}$:

$$
\begin{aligned}
SCD^{trop} =\ & dSCD + SCD_0^{trop} + \Delta SCD^{strat} \\
=\ & dSCD \\
& + VCD_0^{trop} \times AMF_0^{trop} \\
& + VCD_0^{strat} \times AMF_0^{strat} - VCD^{strat} \times AMF^{strat}
\end{aligned}
\tag{2}
$$

The $SCD^{trop}$ is then converted to the desired tropospheric vertical columns ($VCD^{trop}$) through division by the tropospheric air mass factor $AMF^{trop}$:

$$
\begin{aligned}
VCD^{trop} =\ & \frac{SCD^{trop}}{AMF^{trop}} \\
=\ & \frac{dSCD + SCD_0^{trop} + \Delta SCD^{strat}}{AMF^{trop}}
\end{aligned}
\tag{3}
$$



### 4.1.1 Accounting for the tropospheric amount of NO₂ in the background spectrum

In order to correct for the amount of tropospheric $NO_2$ in the background spectrum, $SCD_0^{trop}$ we take an approach similar to the one used by Popp et al. (2012). A tropospheric vertical column of $VCD_0^{trop} = 1 \times 10^{15}\,\mathrm{molec\,cm^{-2}}$ is assumed over the background region, which is a representative value for Europe during the summer period as shown in Huijnen et al. (2010). For each individual measurement during the integration time of the reference background, the AMF is computed for the respective observation geometry, as will be shown in Sec. 4.2, and multiplied with the $VCD_0^{trop}$. The average of the product is used as the tropospheric part of the reference background ($SCD_0^{trop}$).

### 4.1.2 Stratospheric correction

Changes in the stratospheric slant column, as compared to the background scene, propagate to the tropospheric columns measured by AirMAP. Thus we correct for changes in the stratospheric column of $NO_2$, in the term $\Delta SCD^{strat}$. The stratospheric correction is applied using the B3dCTM model. The "Bremen 3d Chemical Transport Model" is a combination of the "Bremen transport model" (Sinnhuber et al., 2003a) with the chemistry code of the "Bremen two-dimensional model of the stratosphere and mesosphere" (Sinnhuber et al., 2003b; Winkler et al., 2008), which evolved from SCLIMCAT (Chipperfield, 1999). The model is driven by ECMWF ERA-Interim meteorological reanalysis fields (Dee et al., 2011). The description of the model setup can be found in Hilboll et al. (2013). The B3dCTM model provides stratospheric VCDs of $NO_2$ on a global grid in a resolution of $2.5° \times 3.75°$. The model can reproduce relative changes in the stratosphere quite well, but biases exist in the absolute amounts of $NO_2$. Thus we do not use the model data directly, but scale the values to match GOME-2 satellite measurements over the clean Pacific at the latitude of Bucharest. The scaling factor $f_{mod}^{sat}$ is derived from the following formula:

$$f_{mod}^{sat} = \frac{VCD_{sat}(Pacific)}{VCD_{B3d}(Pacific)} \tag{4}$$

where $VCD_{sat}$ is the average stratospheric VCD of $NO_2$ in the longitude range 180° - 220° and at the latitude of Bucharest on the day of the measurement. $VCD_{B3d}$ is the modeled VCD in the same region at the time of the satellite overpass.

To correct the change in the stratospheric $NO_2$ over time, the following formula is applied for each measurement, in which the stratospheric AMF is approximated geometrically. The geometric approximation of the stratospheric AMF is valid for Solar Zenith Angles (SZA) < 70° (Burrows et al., 2011, p. 91), and can be applied, because the Solar Zenith Angle was much smaller during the measurements, cf. Table 1. A model value is obtained for each location $(lat, lon)$ and time $(t)$ of the measurements.

$$
\begin{aligned}
SCD^{strat} &= VCD^{strat} \times AMF^{strat} \\
&= f_{mod}^{sat} \times VCD_{B3d}(lat, lon, t) \times \frac{1}{\cos(SZA)}
\end{aligned}
\tag{5}
$$

As the measurements shown in this study were performed around noon, the diurnal variation in $SCD^{strat}$ are very small and could thus be neglected. However, the correction for variations of $SCD^{strat}$ becomes more important for flights performed at large SZA and was therefore implemented in the data processing chain.



## 4.2 Computation of Air Mass Factors

The tropospheric AMF is simulated by a Radiative Transfer Model (RTM). Here the SCIATRAN RTM is used (Rozanov et al., 2014). Table 4 lists the parameters, that are used to calculate AMFs for the individual measurements.

**Table 4.** Input parameters for the calculation of AMFs and the atmospheric correction to derive surface reflectances

| Parameter | Source | AMF | Surface reflectance |
| --- | --- | --- | --- |
| Flight altitude (H) | GPS + correction by DEM | x | x |
| Ground surface reflectance (A) | Intensity + atm. corr. | x | x |
| Viewing zenith angle (VZA) | Calculated | x | x |
| Relative azimuth angle (RAA) | Calculated | x | x |
| Solar zenith angle (SZA) | Calculated | x | x |
| Wavelength ($\lambda$) | Center of fit window | x | x |
| $NO_2$-Profile | Assumption: box-profile in lowest 500 m | x | - |
| Aerosols | INOE Raman-Lidar | x | x |
| | FUBISS-ASA-2 (not coincident) | | |

The angles VAA, RAA and SZA are calulated from GPS position, AHRS and AirMAP's viewing geometry. The gridpoints used in the RTM calculation are listed in the appendix.

The flight altitude, H, is the altitude of the aircraft above ground level. The $NO_2$-profile and the flight altitude affect the

5 sensitivity of the measurements for $NO_2$, because only part of the photons received at the instrument may have passed through atmospheric layers close to the ground. Due to scattering, the measurement sensitivity for the presence of an absorber generally decreases towards the ground, and this effect is more pronounced with increasing flight altitude and small surface reflectances. This is illustrated by Fig. 4, showing the Box-AMF (BAMF) for a typical flight scenario in a Rayleigh atmosphere for two surface reflectances, along with the assumed $NO_2$ box profile.

The BAMF describes the sensitivity of the measurements for an absorber in a certain altitude layer. Almost all photons received at the aircraft have passed the layer just below the flight altitude, thus exhibiting the highest sensitivity to that layer. The ground spectral surface reflectance determines the wavelength dependent fraction of light reflected at the surface. Bright surfaces increase the relative contribution of light reflected by the surface to the signal received at the aircraft, thereby increasing the sensitivity to absorbers located close to the ground. Areas with a high surface reflectance in the fitting window will therefore

generally yield larger dSCDs for the same amount of the trace gas present below the aircraft. This is the reason for the observed small scale structures mentioned in Sec. 3.4. Figure 5 shows the dependence of the AMF on the surface reflectance for the same scenario as described in Fig. 4. The AMF has a strong non-linear dependence on the surface reflectance, especially for dark surfaces. Consequently good knowledge of the surface reflectance is required in order to appropriately correct for its influence on the retrieved trace gas amounts.




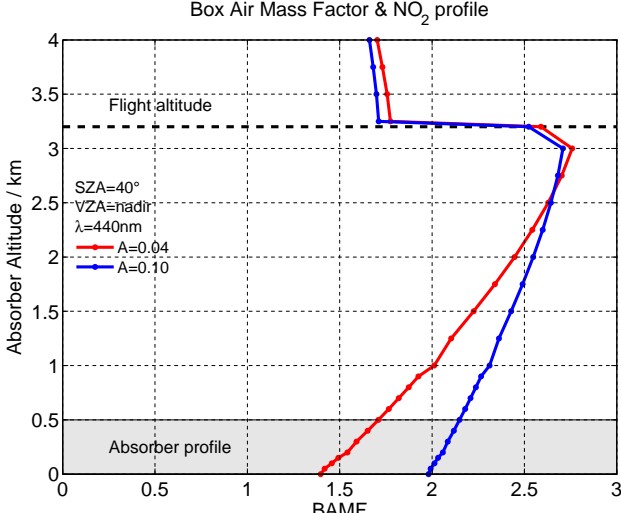

**Figure 4.** Figure showing the Box Air Mass Factor (BAMF) for a flight altitude of 3.2 km, a SZA of 40°, a VZA of 0° at a wavelength of 440 nm for two different surface reflectances of 0.04 and 0.1 in an atmosphere without aerosols. The shaded area shows the assumed box profile of $NO_2$ with a constant mixing ratio in the lowest 500 m.

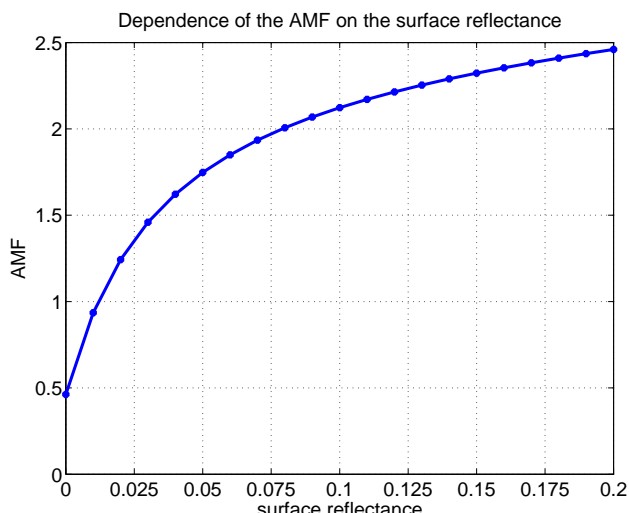

**Figure 5.** Dependence of the tropospheric AMF on the surface reflectance for a flight altitude of 3.2 km, a SZA of 40°, a VZA of 0° at a wavelength of 440 nm in an atmosphere without aerosols.

The observation geometry relevant for the RTM calculations is described by three angles: Solar Zenith Angle(SZA), Viewing Zenith Angle (VZA) and Relative Azimuth Angle (RAA). Figure 6 illustrates the meaning of these angles. The RAA is not shown, but calculated from the difference between the Viewing Azimuth Angle (VAA) and Solar Azimuth Angle (SAA).





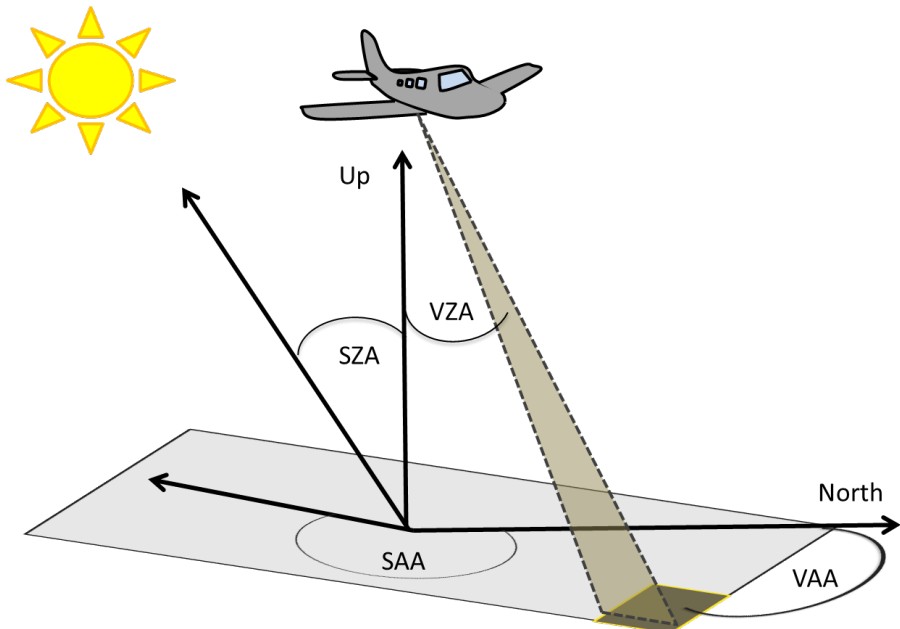

**Figure 6.** Illustration of the angles describing the observation geometry relevant for RTM calculations. The RAA (not shown) is the difference between VAA and SAA.

The VZA is the deviation from the direct nadir observation geometry. As the VZA increases, the light paths get longer. The VZA changes with the viewing direction, but is also altered with the aircraft's attitude. The Relative Azimuth Angle (RAA) is the difference between the Solar Azimuth Angle (SAA) and the Viewing Azimuth Angle (VAA) of the measurement. Following the SCIATRAN convention, the relative azimuth is defined as 0° if the instrument is pointed towards the sun (forward scattering) and 180° for the direction away from the sun (backward scattering). The SZA is the angle between the zenith and the center of the sun's disc and impacts on the length of the light path through the Earth's atmosphere.

The input parameters to SCIATRAN are either measured directly or are calculated from other known parameters, see Table 4. Estimations or assumptions have to be made for the $NO_2$-profile, the aerosol load and the ground surface reflectance. The $NO_2$-profile is assumed to be a simple box-profile with a uniform mixing ratio in the lowest 500 m. For the specific flights, presented here, no direct information on the aerosol profile exist. However, aerosol extinction profiles were measured during the flight at the EARLINET station INOE with a Raman-LiDAR at 532 nm (D'Amico et al., 2015; Mattis et al., 2016; D'Amico et al., 2016). This EARLINET station (www.earlinet.org) is located in Magurele at the outskirts of Bucharest, cf. Fig 1. Furthermore, aerosol profiles were measured with an airborne sun-photometer, the FUBISS-ASA2 instrument (Zieger et al., 2007), in the vicinity of Bucharest during another campaign (AROMAT-2) one year after the flights discussed here. The FUBISS profiles were measured on 30 and 31 August 2015, respectively. The extinction profiles derived from the ground based Raman-LiDAR (INOE) and the airborne sun-photometer (FUBISS) are displayed in Fig. 7. The initially higher vertical resolution of the measured profiles was reduced, by binning to vertical layers of 200 m (INOE) and 240 m (FUBISS). This step was necessary





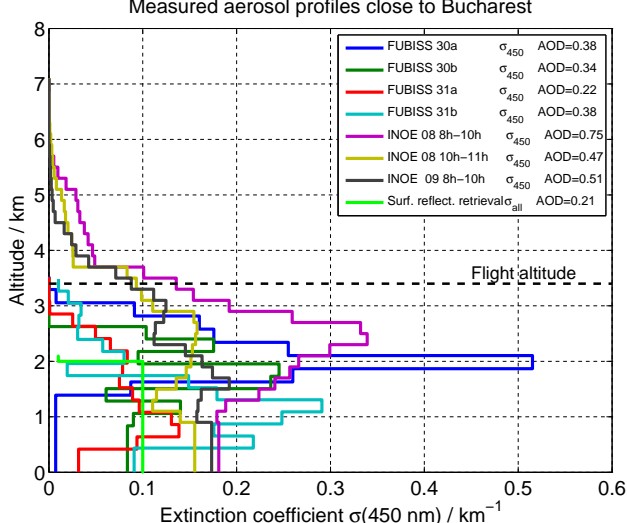

**Figure 7.** Aerosol profiles used in this study. The FUBISS profiles were derived from measurements of the FUBISS-ASA2 instrument in the vicinity of Bucharest at a wavelength of 450 nm. These measurements were performed during the AROMAT-2 campaign one year later. The profiles with the label INOE correspond to profiles derived from ground based Raman-LiDAR measurements at INOE. The legend also shows the respective value of the Aerosol Optical Depth (AOD), which is the integral of the extinction coefficient. In order to allow a better comparison of the AOD values the INOE extinction profiles were only integrated from the ground to the flight altitude.

for the RTM calculations. The names of the profiles indicate the day of measurement. So for example the profile named "INOE 08 8h-10h" corresponds to measurements taken on 2014-09-08 in the time interval 8:00 UTC to 10:00 UTC. The legend also shows the corresponding AOD, which is the integral of the aerosol extinction profile. For better comparability, the INOE profiles were only integrated from the ground to the flight altitude. Because no measurements are available at altitudes close to

the ground, the extinction coefficient of the lowest available altitude was applied to all layers below.

For the analysis of both flights in Sec. 7, we have used the profile FUBISS 31a. The effects of the aerosol assumptions on the AMFs will be discussed in Sec. 8.2.3. Due to the lack of further information, we assume that the aerosol profiles shown in Fig. 7 represent the prevailing aerosol conditions during the flights investigated here. For all aerosol profiles and all atmospheric layers, an asymmetry factor of 0.7 and a Single Scattering Albedo (SSA) of 0.9 was assumed in the RTM calculations, which

is a reasonable choice for urban aerosols (Ricchiazzi et al., 1998). A constant Angstrom exponent of 1.5 was used in order to convert the extinction coefficients to the desired wavelength.

To our knowledge, no data product provides information on the ground spectral surface reflectance in sufficient spatial resolution to be used for our measurements. Such data is only acquired on campaign basis from instruments such as APEX (Itten et al., 2008) or HySpex (Baumgartner et al., 2012). Since the surface reflectance has a large impact on the AMF, cf.

Figure 5, we estimate the surface reflectance from our own measurements. The approach used is described in the following.





## 5   Derivation of surface reflectance

The measured spectra contain information on the recorded light intensity during the exposure. These measured intensities are largely influenced by the surface reflectance. The variability of the measured intensities, however, does not only depend on the surface reflectance, $A$, but also on other parameters such as the observation geometry, the solar position and the aerosol

load. As the AirMAP instrument is not radiometrically calibrated, the measured intensities can be used only in a relative sense. The influence of the observation geometry and the surface reflectance on the radiances measured by AirMAP were modeled for a given aerosol scenario (green line in Fig. 7) with the SCIATRAN RTM and compiled into a look-up-table (LUT). The parameters accounted for in the RTM calculations are listed in Table 4. In order to isolate the contribution of the surface reflectance to the measured intensity, a 6 step procedure (a-f) is applied. The individual steps are explained in the following.

(a) A reference area with a known surface reflectance is identified, that is large enough to contain footprints of several measurements of each viewing direction in that area. The reference area and its surface reflectance value are taken from the ADAM database (A surface reflectance DAtabase for ESA's earth observation Missions) (Prunet et al., 2013). The ADAM database includes a climatology of surface reflectances derived from MODIS (Moderate Resolution Imaging Spectroradiometer) satellite data (MOD09A1 MODIS/Terra Surface Reflectance 8-Day L3 Global 500m SIN Grid V005). It contains normalized surface

reflectances (SZA=45°, VZA=nadir) on a global grid of 0.1° × 0.1° for each month. Unfortunately, the shortest wavelength range, for which surface reflectances are available, without extrapolation, is the 459-479 nm band. The fit window used in our DOAS retrieval is 425-450 nm. However, large differences of the surface reflectance between the two spectral ranges are not expected. Thus, the respective surface reflectance value for the 459-479 nm band is applied here as reference surface reflectance $A_{Ref}$. Two grid cells of the ADAM database were used as the reference region, both having a surface reflectance value $A_{Ref}$

of 0.0394.

The assumption of a small variation between the two spectral bands can be verified by comparison to the OMLER database (Kleipool et al., 2008), a gridded climatology of surface reflectances determined from measurements of the OMI instrument, covering both spectral ranges at a spatial resolution of 0.5°×0.5°. The grid cell closest to the measurement location and time (Latitude=44.5°, Longitude=26°, Month=September, Dataset=MonthlyMinimumSurfaceReflectance) has a mean surface re-

flectance of 0.041 in the 425-450 nm band and a mean surface reflectance of 0.042 in the 459-479 nm band, which corresponds to a relative difference of less than 3 % between the two spectral bands and agrees well with the value from the ADAM database.

(b) As for the $NO_2$ retrieval, the individual viewing directions are treated separately. For each measurement of one viewing direction, with a footprint in the reference area $i$, the intensity is averaged to the value $I_{meas_{Ref}}$. For simplification of the notation, the parameters of the observation geometry are summarized in the parameter set $P$. The flight altitude throughout the

measurements was constant.



$$I_{meas_{Ref}} = \frac{1}{n}\sum_{i=1}^{n} I_{meas_i}(H_i, VZA_i, RAA_i, SZA_i, \lambda) \tag{6}$$

$$= \frac{1}{n}\sum_{i=1}^{n} I_{meas_i}(P_i, \lambda)$$

The measured intensities in the fit window along with the reference region can be seen in Fig 8.

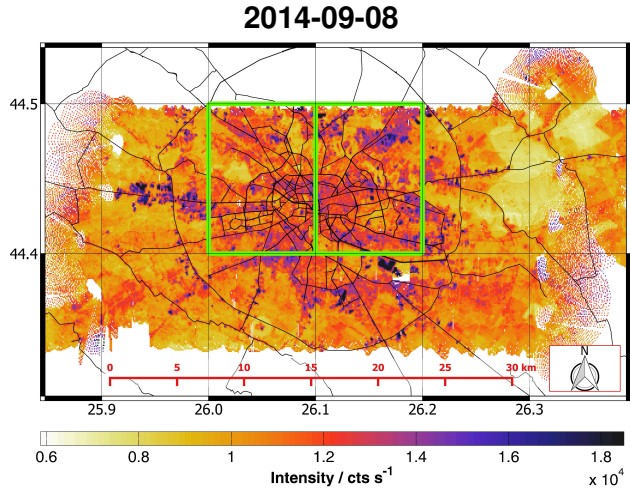

**Figure 8.** Measured intensities on 2014-09-08 along with the reference region from the ADAM database (green boxes).

(c) For each measurement with a footprint in the reference area $i$, the LUT is queried for a modeled intensity at the respective observation geometry and the surface reflectance value of the reference region, using linear interpolation. The wavelength, $\lambda$ was set to the center of the fit window (437.5 nm). These modeled intensities are then averaged to the mean value $I_{mod_{Ref}}$.

$$I_{mod_{Ref}} = \frac{1}{n}\sum_{i=1}^{n} I_{mod_i}(H_i, VZA_i, RAA_i, SZA_i, A_{Ref}, \lambda) \tag{7}$$

$$= \frac{1}{n}\sum_{i=1}^{n} I_{mod_i}(P_i, A_{Ref}, \lambda)$$

(d) In the next step, each measured intensity of the flight, is normalized to match the modeled intensities by scaling with the ratio of modeled and measured intensities above the reference region. This procedure assumes, that the uncalibrated intensities of AirMAP can be calibrated using a single factor, per viewing direction, derived over the reference region.

$$I_{scaled} = I_{meas} \times \frac{I_{mod_{Ref}}}{I_{meas_{Ref}}} \tag{8}$$





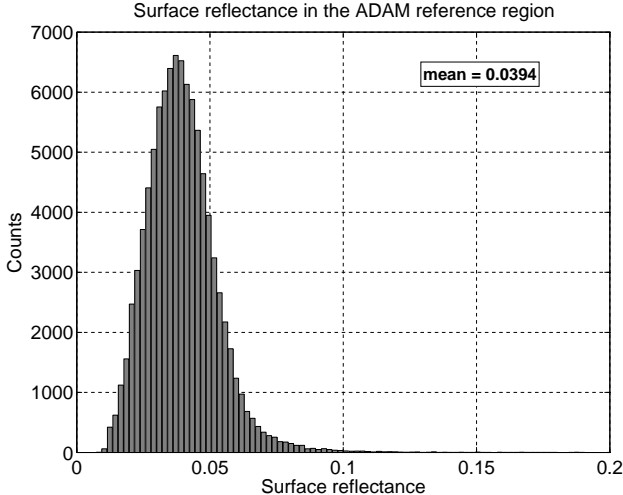

**Figure 9.** Surface reflectances derived from measured intensities for the flight on 2014-09-08 inside the ADAM reference region.

(e) For each measurement, a vector of corresponding modeled intensities for the viewing geometry, $P$, and for all surface reflectances is retrieved from the LUT.

(f) The surface reflectance for the measurement is then determined by selecting the surface reflectance value, for which the modeled intensity from the LUT best fits the scaled measured intensity. In order to improve accuracy, linear interpolation is applied to determine the surface reflectance.

Figure 9 shows a histogram of the derived surface reflectances inside the reference region using the method described. The mean of the surface reflectances agrees with the value of $A_{Ref}$. The values range from 0.005 to about 0.1. Only very few measurements show larger values.

A map of the intensity derived surface reflectances for the flight on 2014-09-08 is displayed in Fig. 10. The smallest surface reflectances are found in the forest and water areas, see map in Fig. 1. The largest values are found above bright rooftops, which is in qualitative agreement with surface reflectances for urban surfaces in the literature (Herold et al., 2004; Heiden et al., 2007).

## 5.1 Surface reflectance data quality

Although the surface reflectance is derived individually for each of the 35 viewing directions, the surface reflectances show a consistent behavior. This can be seen over areas with a homogeneous surface type, such as the large forest area in the North-East of the flight pattern. Although the surface reflectances were derived for all viewing directions independently, the homogeneous surface type also yields a homogeneous surface reflectance.

To investigate further on the precision of the method, the derived surface reflectances of two flights on different days were spatially binned on the same grid with a resolution of $0.0008° \times 0.0008°$. A pixel-wise comparison is shown in Fig. 11 (a).





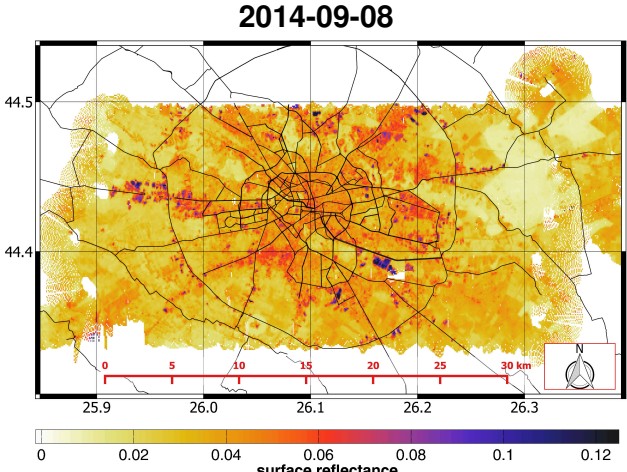

**Figure 10.** Surface reflectances for the Bucharest region derived from measured intensities for the flight on 2014-09-08

The derived values of the two flights agree well having a correlation coefficient of R=0.91 and a slope close to one. Figure 11 (b) shows the corresponding histogram of absolute differences. The mean of the differences is close to zero and the FWHM (Full Width Half Maximum) of the differences is less than 0.01. These results indicate that the applied corrections for viewing geometry and atmospheric effects are reasonably consistent.

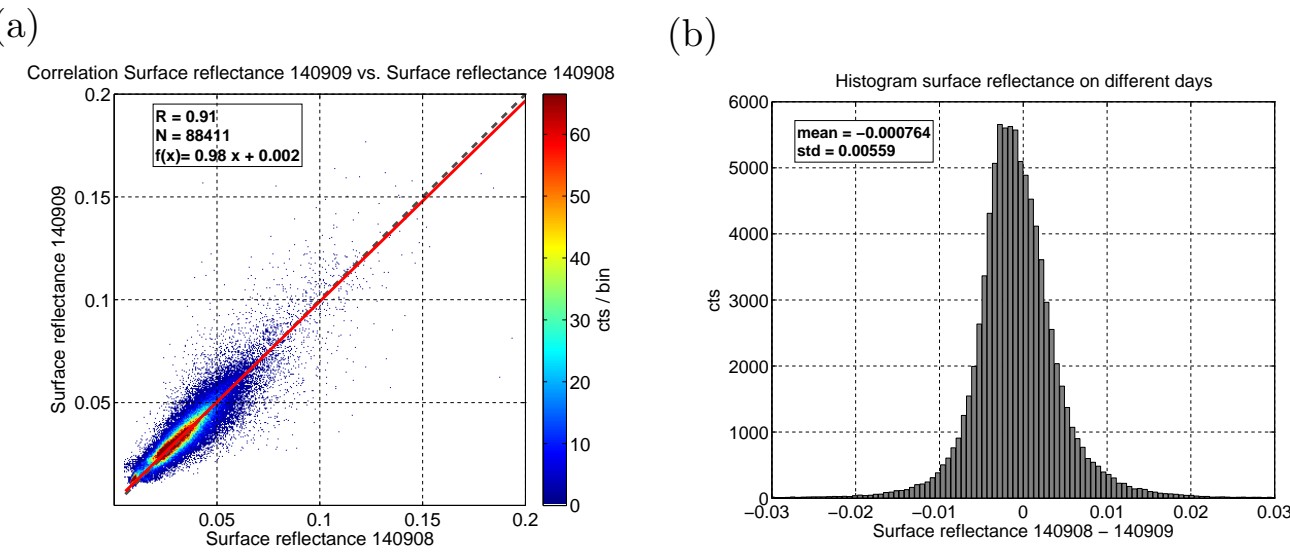

**Figure 11.** Pixel-wise comparison of co-located surface reflectances, derived from flights on different days, 2014-09-08 and 2014-09-09. Panel (a) Shows the correlation between the two flights. The text box shows the correlation coefficient R, the number of data points N, and the result of a linear orthogonal fit (red line). Panel (b) shows the corresponding histogram of the absolute differences.





Another and similar approach to test the precision of the method is to investigate the derived surface reflectances of multiple overpasses over the same location during one flight. A large fraction of the flight area was covered twice, due to an intended overlap between adjacent tracks. The individual tracks were flown in East-West / West-East direction around noon. This implies, that approximately half of the sensor swath is pointing towards the sun ($0° <$ RAA $< 90°$), while the other half is pointing away from the sun ($90° <$ RAA $< 180°$). Figure 12(a) shows a correlation plot of the derived surface reflectances from the flight on 2014-09-08 in dependence of the RAA. Figure 12(b) shows the corresponding histogram of the absolute differences. It can be

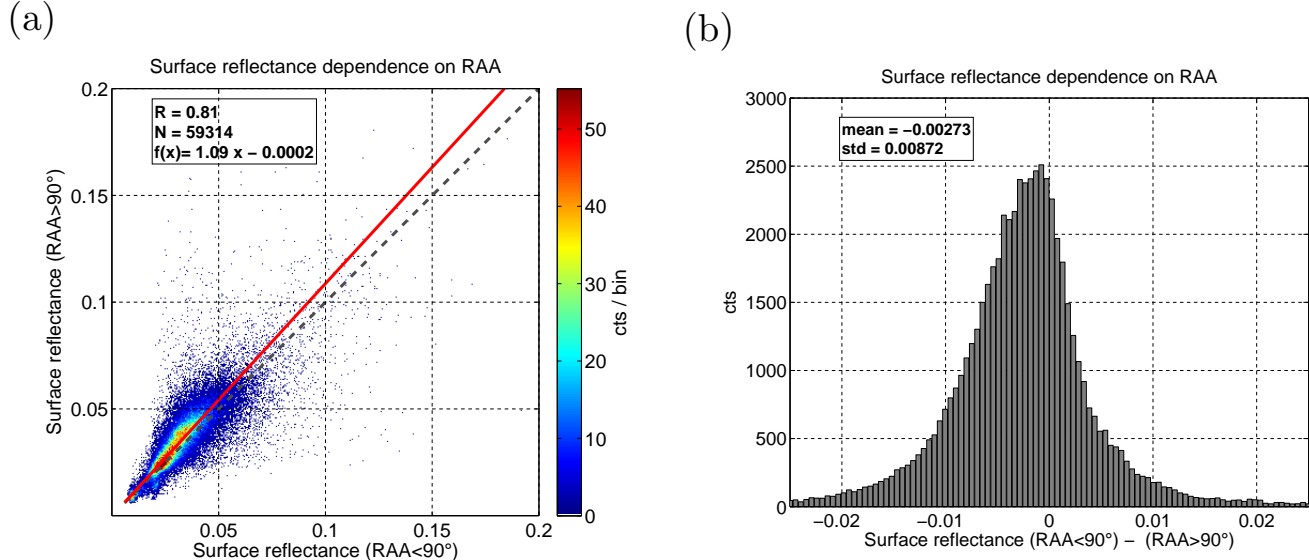

**Figure 12.** Surface reflectance in dependence of the Relative Azimuth Angle (RAA), as determined from one flight on 2014-09-08. Panel (a) shows the correlation of surface reflectances acquired under forward and backward scattering regimes. The text box shows the correlation coefficient R, the number of data points N, and the result of a linear orthogonal fit (red line). Panel (b) shows the corresponding histogram of absolute differences.

seen, that the surface reflectances acquired under a RAA $< 90°$ (pointing towards sun) generally yield lower values.

The spatial distribution of these differences is displayed in Fig. 13. The largest differences occur in the urban most densely populated and built-up areas and over water bodies, which can be clearly identified, cf. Fig. 1. Regions covered by vegetation, such as the forest in the North-East or the regions without buildings show much less differences. The reason for this behavior is attributed to the angular dependency of the ground surface reflection, as represented by the BRDF (Bidirectional Reflectance Distribution Function). This can be illustrated by a simple example. In the urban area, usually the reflecting surface is not flat, but rather structured. Houses will appear brighter on the side facing the sun, while they appear darker on the opposite side. In general, flat surfaces such as lakes appear brighter when observed at RAA$<90°$ and darker at RAA$>90°$, while the opposite is true for non-flat surfaces such as woodlands. Full consideration of these effects is out of scope of this study. Missing treatment of BRDF effects introduces a relative uncertainty on the surface reflectance of around 10 % as can be seen from Fig. 12.



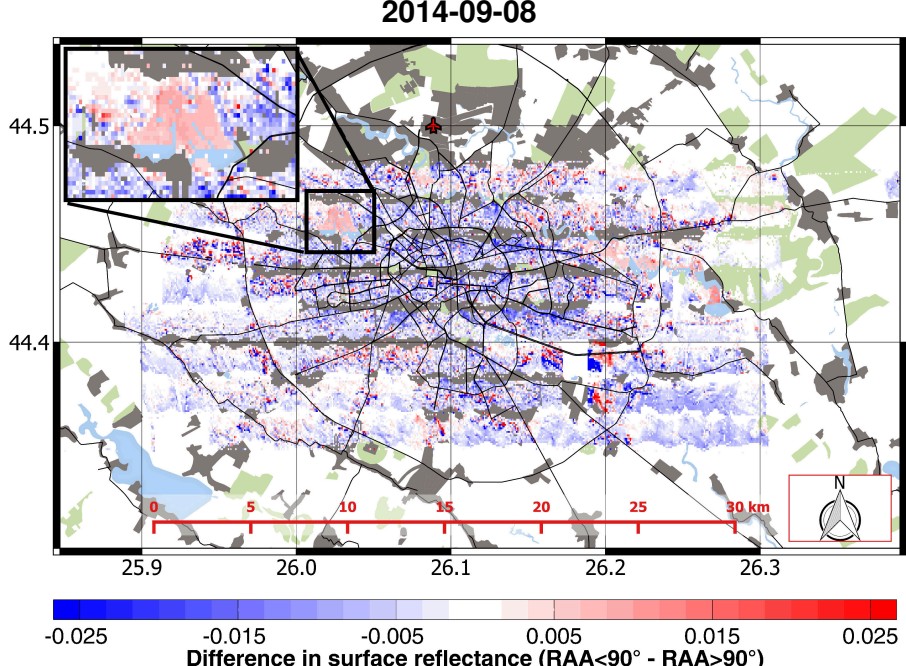

**Figure 13.** Absolute differences of surface reflectance in dependence of the Relative Azimuth Angle (RAA), as determined from one flight on 2014-08-08. Additionally shown is a zoom-in on the lake, highlighting the differences observed over water bodies. The base map of Bucharest is shown in the background.

However, when applying the intensity derived surface reflectance to the LUT of AMFs, discarding these directional effects has only a minor impact on the retrieval. The important information for a correct AMF, matching the scene, is the amount of light that was reflected on the ground. Since each spectrum in the DOAS analysis has its own surface reflectance value, recorded under the same observation geometry, the AMF is corrected implicitly. The derived surface reflectances can thus be regarded 5   as effective reflectance values.





## 6 Effects of the applied surface reflectance on the results

Figure 14 shows a zoom-in on the map with the highest pollution levels during the flight on 2014-09-08. When comparing the measured intensity (a) and the dSCDs (c), the spatial correlation of bright surfaces with high dSCDs, as mentioned in Sec. 3.4, is clearly visible. The AMF, (b), is dominated by the amount of light reflected by the surface. This was accounted for in the

5  AMF by the application of the intensity derived surface reflectances. The VCDs in (d) are much smoother than the dSCDs. Small scale structures, originating mainly from the surface reflectance are successfully eliminated.

**(a)   Intensity**

**(b)   AMF**

**(c)   dSCD**

**(d)   VCD**

**Figure 14.** Overview of the Intensity, AMF, dSCD and VCD for the flight on 2014-09-08. Application of the intensity derived surface reflectances generates a smooth $NO_2$ VCD distribution.





## 7 NO$_2$ VCD above Bucharest

Figure 15 and Fig. 16 show the VCD of NO$_2$ retrieved from the flights on 2014-09-08 and 2014-09-09, respectively. Both figures show bin averaged VCD values on a regular grid with a spatial resolution of $0.0008° \times 0.0008°$. Individual measurements with a flight altitude lower than 3000 m or a large fitting error (RMS larger than 0.02) or a VZA lager 40° were filtered out
prior to the gridding procedure.

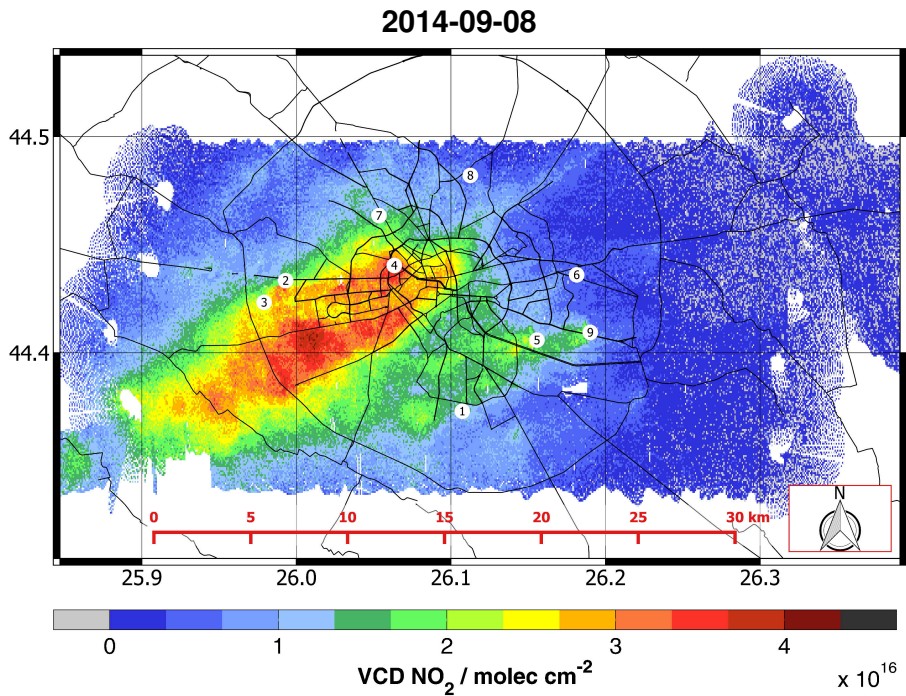

**Figure 15.** Vertical column densities measured on 2014-09-08. The numbered labels show NO$_x$ emitters listed in the E-PRTR, cf. Table 5.

The numbered labels show NO$_x$ emitters that are listed in the European Pollutant Release and Transfer Register, E-PRTR, (European Environment Agency). Further details about these pollution sources are given in Table 5.

Besides the large plume above the city, several emission hot spots are identified, which correlate well with the locations of the facilities listed in the E-PRTR (e.g. 5, 9 in Fig. 15 and 2, 3 in Fig. 16). It should be noted, that not all of the listed emitters have
necessarily been active during the time of the research flight. Despite the fact, that both flights were performed under similar conditions, (Monday / Tuesday, around local noon, similar wind speed), the horizontal NO$_2$ distribution is quite different. The flight on 2014-09-08 has larger NO$_2$ VCD of up to $4.2 \times 10^{16}$ molec cm$^{-2}$, while the maximum NO$_2$ VCD for the flight on 2014-09-09 is $3.4 \times 10^{16}$ molec cm$^{-2}$. It should be noted, that despite the different appearances and peak values, the mean NO$_2$ amount of both flights are similar: The average NO$_2$ VCD is $9.3 \times 10^{15}$ molec cm$^{-2}$ on 2014-09-08 and $8.9 \times 10^{15}$ molec cm$^{-2}$
on 2014-09-09.



**Figure 16.** Vertical column densities measured on 2014-09-09. The numbered labels show $NO_x$ emitters listed in the E-PRTR, cf. Table 5.

**Table 5.** $NO_x$ Emitters listed in the European Pollutant Release and Transfer Register (E-PRTR)

| # | $NO_x$ Emission [kg yr$^{-1}$] | Reporting Year | Facility ID |
|---|---|---|---|
| 1 | 128000 | 2014 | 167470 |
| 2 | 122000 | 2014 | 193189 |
| 3 | 433000 | 2013 | 167472 |
| 4 | 266000 | 2013 | 167473 |
| 5 | 994000 | 2012 | 167471 |
| 6 | 140000 | 2007 | 22960 |
| 7 | 165000 | 2007 | 22961 |
| 8 | 161000 | 2007 | 167474 |
| 9 | 144000 | 2007 | 22964 |

Compare entries with locations in Fig. 15 and Fig. 16. The E-PRTR lists $NO_x$
emitters exceeding a threshold of $1 \times 10^5$ kg yr$^{-1}$.

The reason for the different $NO_2$ distribution is not completely understood but is probably attributed to the wind conditions. Figure 17 shows the wind properties for the two investigated days and one day before, measured at Baneasa Airport in the North of Bucharest, cf. Fig. 1. The shaded areas indicate the times of the measurements.




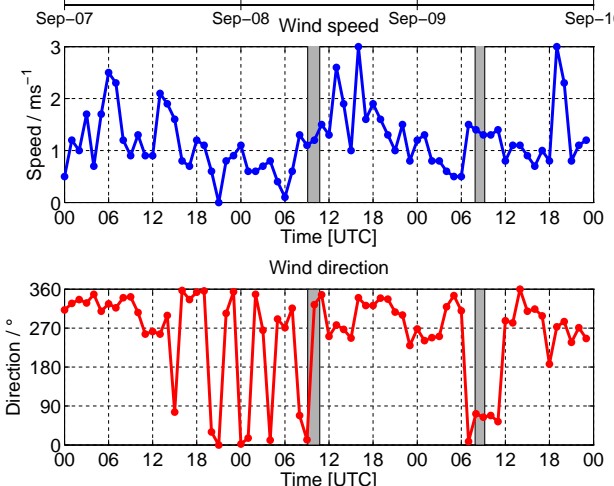

**Figure 17.** Wind data for the two investigated days and the day before measured at the Baneasa Airport. The gray shaded area indicates the times of the flights. Data provided by Meteo Romania.

On both days there was a similar low wind speed from Northern or North-East-East directions, respectively, during the time of the measurements. Before the first flight however, there was actually no significant wind at all during the morning rush-hour. This results in a stronger accumulation of $NO_2$ close to the sources. If this is the case, it is possible, that the $NO_2$ could reach higher altitudes before being transported away. This results in a higher sensitivity towards the $NO_2$ because of a different AMF, see Sec. 4.2. Assuming a wind speed of $1.4\,\mathrm{ms}^{-1}$, an air parcel would be transported $5\,\mathrm{km}$ per hour. The measured plumes extend to about $15\,\mathrm{km}$ from the city center, where the densest traffic is expected. This means, that the observed $NO_2$ could already be up to $3\,\mathrm{h}$ old. However, from the available data, no unambiguous and firm conclusion can be drawn. To investigate on this issue further, reliable high resolution data describing the meteorological conditions and emissions of $NO_x$ would be necessary. Because of the very slow wind speed, uncertainties on the transport of $NO_2$ are relatively large.

The area covered during a research flight corresponds to almost two OMI pixels or 23 S5p pixels. Averaging the AirMAP measurements could thus be used for satellite data validation. We do not show a comparison to OMI satellite data in this study, because on the investigated days, the OMI footprints did not match well with the covered area. Considering the large spatial gradients of the $NO_2$ field, a comparison of only partially overlapping measurement areas is not very meaningful. However, a comparison to mobile car-DOAS measurements will be shown in Sec. 9.

# 8   Discussion of uncertainties

The total uncertainty on the vertical column originates from (1) uncertainties on the retrieved dSCDs (2) uncertainties in the applied AMFs and (3) uncertainties in the background column. The contribution of the different uncertainties on the final VCD result are discussed in the following.





## 8.1 Uncertainty on the differential slant column densities

Several effects contribute to the dSCD uncertainty: Shot noise from the radiance, electronic noise from the instrument, uncertainties from the cross-sections (typically around 2-3 %), and errors from spectral interferences in the DOAS retrieval. The resulting individual relative fitting errors of $NO_2$ above the polluted city center of Bucharest range from 5 % to 10 %. For smaller $NO_2$ abundances, the relative error is much larger as some of the error sources are absolute errors that do not scale with the $NO_2$ signal. Therefore a relative as well as an absolute error needs to be stated. The combined uncertainty of the dSCDs is then the sum of the relative and the absolute errors. The random error of the dSCDs can be estimated from the noise of the retrieved dSCDs in the time and region where the background spectrum was taken (Schönhardt et al., 2015; Platt and Stutz, 2008, chap. 8). Provided that the tropospheric $NO_2$ column in that area is small and constant, the observations are scattered around zero, and the RMSE (root-mean-squared error) provides an estimate on the magnitude of the random errors. Due to the variation in spectral resolution, the RMSE of the dSCDs varies with the viewing direction, and ranges from $2.1 \times 10^{15} \, \mathrm{molec \, cm^{-2}}$ in the central viewing directions to $2.7 \times 10^{15} \, \mathrm{molec \, cm^{-2}}$ in the outer viewing directions. The mean dSCD error for the flight on 2014-09-08, as output of the DOAS fit, is $2.2 \times 10^{15} \, \mathrm{molec \, cm^{-2}}$, which is in agreement with the RMSE in the background $I_0$-region.

## 8.2 Uncertainties on the air mass factors

The AMF converts the dSCDs to VCDs. Thus, uncertainties on the AMF will affect the uncertainties on the VCDs directly, mainly in the form of relative errors. The largest uncertainties contributing to the error on the AMF are the aerosol effects, followed by the surface reflectance and the unknown $NO_2$ profile.

### 8.2.1 Uncertainties on the surface reflectance

Figure 5 shows the strong non-linear dependence of the AMF on the surface reflectance for a typical observation scenario. While the varying surface reflectances are captured well in our retrieved surface reflectances, they all depend on the surface reflectance value of the reference area. Missing treatment of BRDF effects might also cause an uncertainty of around 10% on the retrieved surface reflectance itself. For the application in the AMF-LUT however, the AMFs are implicitly corrected, as discussed in Sec. 5.1.

From the comparison of the surface reflectance derived on two different days in Fig. 11 (b) the precision of the surface reflectance retrieval was assessed to be 0.006. Assuming a surface reflectance of 0.04, this results in a statistical uncertainty of 6 % on the AMF. The value of the precision on the surface reflectance describes an upper limit, because the value determined is influenced by gridding artifacts and includes variations caused by directional reflectance properties of the surface. The effect of aerosols on the surface reflectance retrieval is discussed in Sec. 8.2.3.



### 8.2.2 Uncertainties introduced by the NO$_2$-profile assumptions

Figure 4 shows altitude dependent Box Air Mass Factors (BAMF) for a flight altitude of 3.2 km, a constant surface reflectance of 5% and a solar zenith angle of 40°. The BAMF describes the sensitivity of the retrieved slant column to the presence of a given amount of an absorber like NO$_2$ in a given altitude. From the figure the influence of the absorber profile on the AMF

can be estimated. For the displayed scenario, a surface reflectance of 0.04 and a well-mixed NO$_2$ box profile of 0.5 km height, the resulting AMF would be 1.7. If the maximum profile height is increased to 1 km, the resulting AMF would be 2. Under the assumption, that the profile altitude is in the range of 500 m to 1000 m the uncertainty of the profile on the AMF is on the order of 10 %. The magnitude of the uncertainty introduced in the AMF by the profile uncertainty increases with decreasing albedo and with increasing SZA and can be of the order of 20% or larger for more deviating profiles at lower sun.

### 8.2.3 Uncertainties related to aerosols

Aerosols can have several impacts on the retrieved vertical columns. If a layer of aerosols is present above a trace gas it obscures the view on the trace gas layer by shielding it through the increased scattering probability. This effect would bias the VCD low if not taken into account. On the other hand, aerosols can lead to multiple scattering effects which extend the light path within the aerosol layer. If the aerosols and the trace gas are present in the same layer, this will lead to a larger absorption of the

15 trace gas of interest, biasing the VCD high. These considerations assume aerosols with large single scattering albedo (SSA). For absorbing aerosols, the light path enhancement effect is reduced and VCDs are low biased also in case of a well-mixed trace gas and aerosol layer. Studies for satellite observations (Leitão et al., 2010) report on sensitivities varying between a few percent and up to 20 % for aerosol layers located above the trace gas of interest.

Figure 7 shows aerosol profiles used in the SCIATRAN RTM calculations, which were derived from ground based Raman-

20 LiDAR and airborne sun photometer measurements. The profile used in the analysis (FUBISS 31a) represents a scenario with a rather low aerosol load. Furthermore, the used aerosol profile assumes a small extinction in the lowest 500 m, whereas probably the NO$_2$ mainly resides in that layer. Thus, the examined aerosol profiles mainly reflect the shielding effect of aerosols. Light path enhancements in the trace gas layer by aerosols are thus not represented well. Therefore, the AMFs are probably underestimated and the VCDs are more likely biased high. To study the influence of the aerosol profile on the AMF,

all available profiles were used for AMF calculations and the results compared to the profile used in this study. Figure 18 compares AMFs of the other aerosol scenarios shown in Fig. 7 and AMFs in a pure Rayleigh atmosphere to the profile applied in this study. The shaded area describes a ±30% deviation of the AMFs depending on the used aerosol profile. In the case of a Rayleigh atmosphere, the AMF is larger, because no shielding aerosol effect is present. With increasing extinction above the trace gas layer, the AMF becomes smaller.

Certainly, a single profile cannot describe the aerosol distribution across the whole extent of the investigated area. The INOE profiles were measured at the outskirts of Bucharest. Due to restrictions on the air space, the profiles derived from the FUBISS-ASA2 instrument were also measured outside the city and are not coincident in time. The uncertainties thereby are difficult to quantify. Nevertheless, the range between the investigated aerosol scenarios provides an estimate on the impact of the aerosol





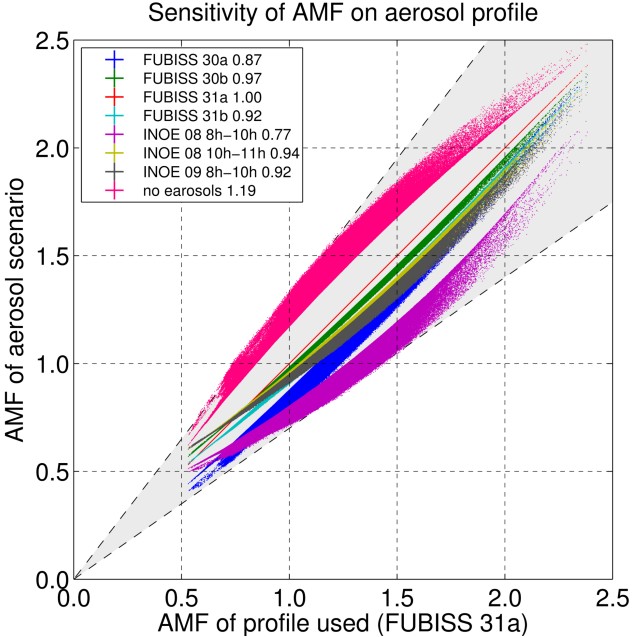

**Figure 18.** Influence of the assumed aerosol profile on the AMFs for an $NO_2$ box profile in the lowest 500m. The shaded area represents a $\pm 30\%$ uncertainty on AMFs of the used aerosol profile. The legend also shows the mean value of the individual ratios between the AMF of the respective scenario to the scenario used.

profile on our measurements.The range of AMFs obtained when using the different aerosol profile assumptions supports the uncertainty values reported in Leitão et al. (2010). It should be noted that the aerosol load also influences the derived surface reflectances. If the true aerosol load is larger, than assumed in the RTM calculations of the modeled intensities, the scaling factor in Eq. 8 becomes smaller, resulting in a low biased surface reflectance. Underestimating the surface reflectance results in

5  a in a low biased AMF and consequently high biased VCDs. This effect of an underestimation of the aerosol load in the RTM calculations for the surface reflectance is antagonistic to the effect on the AMF, because an underestimation of the aerosol in the modeled AMF underestimates the shielding effect of aerosols. As discussed above, an underestimation of the shielding effect causes high biased AMFs, resulting in low biased VCD. It can thus be argued, that these antagonistic effects partly compensate each other. The surface reflectance was derived by using a single simplified aerosol scenario with a low AOD. Applying AMFs

10  corresponding to a a larger aerosol load than accounted for in the surface reflectance retrieval leads to low biased AMFs.





### 8.3 Uncertainty resulting from the $NO_2$ amount in the background spectrum

#### 8.3.1 Uncertainties in the tropospheric background

As no direct measurements of the $NO_2$ column in the background scene ($VCD_0^{trop}$) exists, this value is quite uncertain in relative terms. Assuming a 100% uncertainty on the $1 \times 10^{15}\,\mathrm{molec\,cm^{-2}}$ value, a tropospheric AMF of 1.2 and a tropospheric

AMF over the background scene of 0.8, this adds an uncertainty of $7 \times 10^{14}\,\mathrm{molec\,cm^{-2}}$ to the tropospheric vertical column.

#### 8.3.2 Uncertainties related to changes in stratospheric $NO_2$

The stratospheric $NO_2$ signal changes with the SZA and by photochemical reactions. Our measurements were performed around local noon. Thus the relative changes in the SZA and the light intensity are rather small. Furthermore, we apply a correction for changes in the stratospheric $NO_2$ amount with respect to the background spectrum. The maximum of the change

in the stratospheric SCD is $2.8 \times 10^{14}\,\mathrm{molec\,cm^{-2}}$. Assuming a 100% uncertainty on the applied correction, and a tropospheric AMF of 1.2, the uncertainty on the stratospheric contribution is around $3 \times 10^{14}\,\mathrm{molec\,cm^{-2}}$.

### 8.4 Summary of uncertainties

Table 6 summarizes the major uncertainties deduced from the considerations made above. Assuming that the sources of the uncertainties are unrelated, the combined uncertainty on the AMF can be estimated by the square-root of the quadratic sum of

individual uncertainties. The largest uncertainty on the AMF arises from the assumptions on the aerosol load and properties, followed by the $NO_2$ profile and the surface reflectance. The resulting combined uncertainty on the AMF is less than 26 %. The assumtion of independent uncertainties is not completely valid, because of the link between the retrieved surface reflectance and the shielding effect, which depends on the aerosol load. Because the uncertainty on the dSCD is an absolute value, it does not scale with the $NO_2$ signal. A relative uncertainty is stated for typical dSCD value. Taking the mean dSCD value and the

mean dSCD error as a reference for a typical value, and taking into account the main influences listed in Table 6, the overall uncertainty on the $NO_2$ VCD is about 34 %.

**Table 6.** Major contributors to uncertainties

| Parameter | Uncertainty assumption | Reference case | Relative uncertainty AMF / dSCD | Remark |
|---|---|---|---|---|
| Surface reflectance | 0.006 | 0.04 | 6 % | cf. Fig. 11(b) |
| $NO_2$-box-profile height | +500 m | 500 m | 10 % | cf. Fig. 4 |
| Aerosols | all aerosol scenarios | FUBISS 31a | 3-23 % | cf. Fig. 18 |
| dSCD error | $2.2 \times 10^{15}\,\mathrm{molec\,cm^{-2}}$ | $1 \times 10^{16}\,\mathrm{molec\,cm^{-2}}$ | 22 % | From mean dSCD error & mean dSCD |





## 9 Comparison to mobile car-DOAS measurements

On 2014-09-08 mobile car-DOAS measurements were performed by the Max-Planck Institute for Chemistry in Mainz (MPIC), and the University of Galati (UGAL). For the flight on the next day, no supporting measurements are available because the car-DOAS instruments were transfered to the Turceni power plant, the other campaign measurement site. The most important

properties of the mobile car-DOAS measurement are shown in Table 7. More detailed information about the MPIC and UGAL VCD retrieval algorithms used can be found in Wagner et al. (2010); Shaiganfar et al. (2011) and in Constantin et al. (2013), respectively.

**Table 7.** Properties of the mobile car-DOAS measurements

| Parameter | MPIC | UGAL |
|---|---|---|
| Elevation angle ($\alpha$) | 22° | zenith |
| Fit window | 400-448 nm | 425-490 nm |
| Gaps | 429.7-431.5 nm | - |
| Polynomial degree | 5 | 5 |
| Trace gases | $NO_2$, $O_3$, $O_4$, $H_2O$, Ring | $NO_2$(298/220 K), $O_3$, $O_4$, $H_2O$, Ring |
| AMF calculation | geometric approx. | uvspec/DISORT RTM |
| AMF uncertainty | typ. <20% | typ. <20% |

The data provided by the two groups can be used to validate the VCD retrieved from AirMAP to independent ground measurements. For the comparison, the car-DOAS VCD data was filtered to contain only measurements during the research

flight and gridded to a resolution of 0.03°×0.03°. The same grid was applied to the AirMAP VCDs and a pixel-wise comparison of the co-located pixels was performed. For an overview of the locations of the car measurements refer to Fig. 1. Figure 19 (a) and Fig. 20 (a) show correlation plots, with the VCD retrieved by AirMAP on the x-axis and the VCD retrieved by the MPIC and UGAL car-DOAS instrument on the y-axis. Figure 19 (b) and Fig. 20 (b) show a time series for the car measurements, along with AirMAP's retrieved VCDs at the respective car positions. In this sense, the time axis is only valid for the car-DOAS

measurements. The lower panel shows the temporal difference between the airborne and the ground-based measurements.

Both comparisons reveal a good correlation between the datasets with correlation coefficients of R=0.85 (MPIC) and R=0.94 (UGAL), respectively. The larger spread in the comparison to the MPIC instrument is probably caused by the viewing geometry, the driven route and the spatial pattern and temporal variability of the $NO_2$ field. The spatial information on the location of the car measurements is taken from the car's position during the measurement. The location of the AirMAP measurement is the

center of the projected footprint of the ground pixel. Since the MPIC instrument is pointing at an elevation angle $\alpha$ of 22°, it integrates a different horizontal air mass.



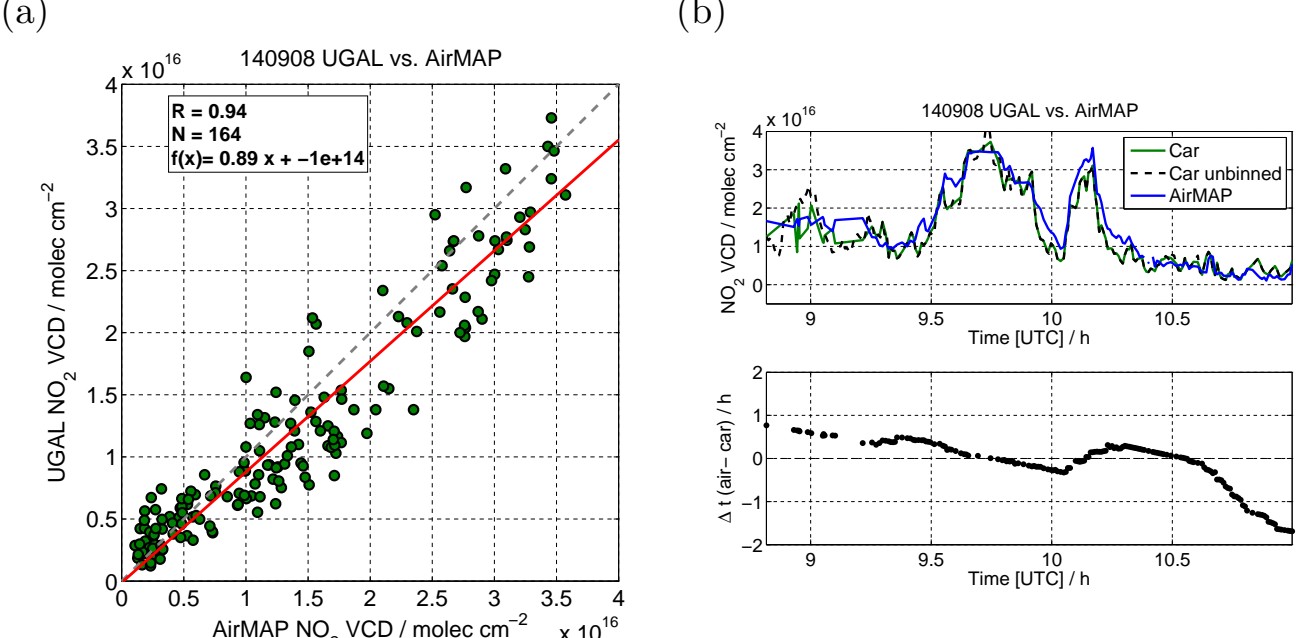

**Figure 19.** Panel (a): Correlation between VCD retrieved by AirMAP and the VCD retrieved by the UGAL car-DOAS instrument measured on 2014-09-08. The text box shows the correlation coefficient R, the number of data points N, and the result of a linear orthogonal fit (red line). Panel (b): Time series of the car-DOAS measurements along with AirMAP measurement at the respective car positions. The lower panel shows the temporal difference between the measurements compared.

When comparing to the UGAL instrument the scatter is much more confined, presumably because the spatial inhomogeneities in the $NO_2$ field affect the comparison to a much smaller extent because the instrument is pointed to the zenith.

The maximal horizontal mismatch $d_{max}$ between the airborne and the ground-based measurement can be approximated by a simple geometrical approach.

$$d_{max} = \frac{H_{NO_2}}{\tan(\alpha)} + H_{NO_2} \times \tan(VZA) \tag{9}$$

For the assumed 500 m $NO_2$-box-profile, $H_{NO_2}$, and a maximum VZA of AirMAP during a level flight, this results in a maximal displacement of 240 m for the zenith pointing UGAL instrument, whereas the possible mismatch for the MPIC instrument adds up to 1480 m. This explanation is further supported by Fig. 20 (b), where an obvious time lag between the two datasets can be seen shortly before 10:00 UTC, which translates to a spatial mismatch.

A linear orthogonal fit to the data reveals a slope of 0.89 for the comparison to the UGAL data, indicating an overestimation of AirMAP's VCDs or an underestimation of the UGAL data. However, the slope in the comparison to the MPIC data has a value of 1. A possible reason for the lower values obtained from the UGAL instrument could be related to the assumptions made on the $NO_2$ profile. To investigate on this hypothesis, the BAMF for ground based zenith sky observations at a SZA





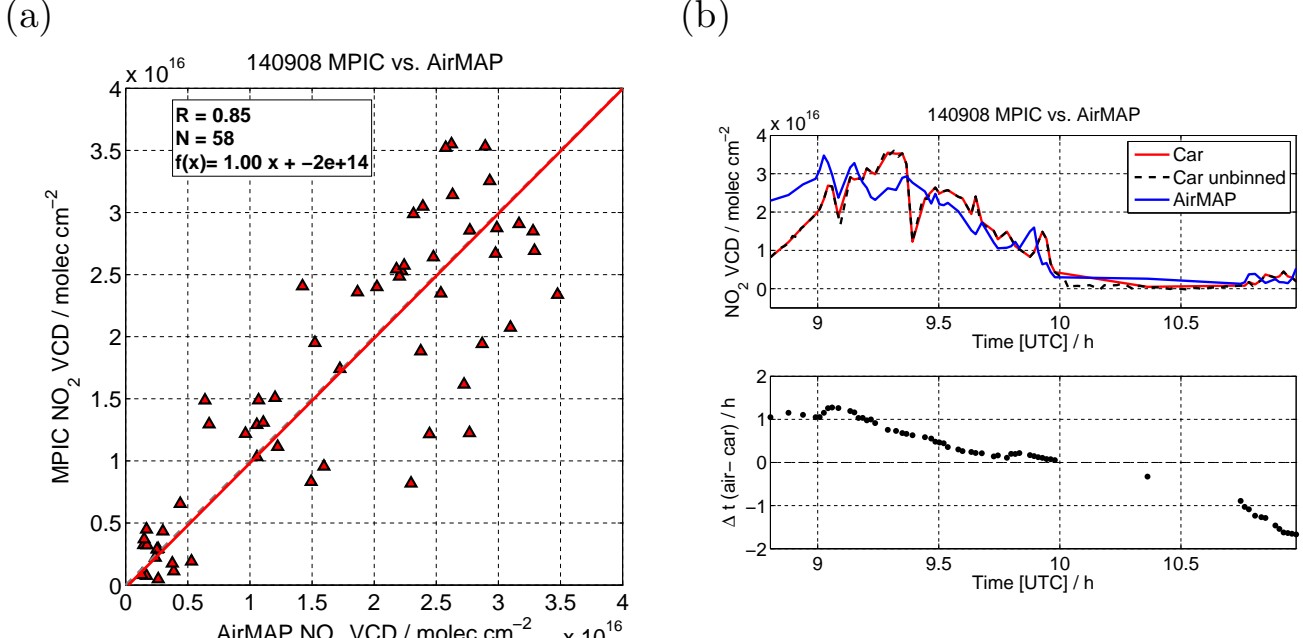

**Figure 20.** Panel (a): Correlation between VCD retrieved by AirMAP and the VCD retrieved by the MPIC car-DOAS instrument measured on 2014-09-08. The text box shows the correlation coefficient R, the number of data points N, and the result of a linear orthogonal fit (red line). Panel (b): Time series of the car-DOAS measurements along with AirMAP measurement at the respective car positions. The lower panel shows the temporal difference between the measurements compared.

of $40°$ was weighted with a) the $500\,\mathrm{m}$ box profile assumed in the AirMAP retrieval and b) the profile used in the UGAL retrieval to yield the respective AMFs. The $NO_2$ profile used by UGAL was extracted from the CHIMERE model for a small town (Timisora, Romania, 300000 inhabitants) and assumes an exponentially decreasing mixing ratio up to an altitude of $6\,\mathrm{km}$. The ratio of these differently weighted BAMFs (AMFs) is 0.90 ($AMF_{500m}/AMF_{UGAL}$). This ratio is close to the slope determined from the fit shown in Fig. 19(a), indicating that the differences between the two instruments can be explained by the assumptions made on the profile. The data retrieved from the different instruments and used in this study was analyzed independently by all groups without common assumptions on the $NO_2$ profile, aerosols and other properties related to the radiative transfer. Considering the large sensitivity of the AirMAP AMFs on the surface reflectance and the aerosol profile, the datasets show good agreement.

## 10 Estimation of the urban $NO_x$ emission rate

Several studies have investigated the $NO_x$ emission rate of point sources and urban areas, e. g. (Ibrahim et al., 2010; Shaiganfar et al., 2011; Beirle et al., 2011; Liu et al., 2016). Here we adapt the method presented in Ibrahim et al. (2010) and Shaiganfar et al. (2011), where urban emissions are estimated from an encircled area, by integrating along the route of a circle $S$. This




method is based on Gauss's divergence theorem, describing the relation between the flux of a vector field through a closed surface (measured) to the divergence of the vector field inside the enclosed volume (emissions inside that volume). The $NO_2$ emission rate $\boldsymbol{F_{NO_2}}$ may be estimated from:

$$\boldsymbol{F_{NO_2}} = \oint_S VCD_{NO_2}(\boldsymbol{s}) \cdot \bar{\boldsymbol{w}} \cdot \bar{\boldsymbol{n}} \cdot \boldsymbol{ds} \tag{10}$$

Here, $\bar{\boldsymbol{n}}$ indicates the normal vector parallel to the Earth's surface and orthogonal to the azimuth of the line segment $\boldsymbol{ds}$ and $\bar{\boldsymbol{w}}$ is the mean horizontal wind vector. $NO_x$ is primarily emitted as NO and converts to $NO_2$ by reaction with $O_3$. $NO_x$ has a short lifetime in the order of hours. In order to derive the $NO_x$ emission rate from the measured $VCD_{NO_2}$, Eq.10 has to be modified to account for a) the partitioning between NO and $NO_2$ and b) chemical loss. Assuming steady state conditions, the partitioning between NO and $NO_2$ can be described by the Leighton relationship. For typical urban conditions at noontime,

the ratio of $[NO]/[NO_2]$ is about 0.32 (Seinfeld and Pandis, 2006). Therefore, the factor $c_L$ =1.32 is introduced, which scales the measured $NO_2$ to $NO_x$. The correction factor $c_\tau$ accounts for chemical loss that occurred from the location of emission to the location of the measurement. It can be estimated from the $NO_x$ lifetime $\tau$, the wind speed $\bar{w}$ and the distance between source and measurement $d$.

$$c_\tau = \exp\left(\frac{d/\bar{w}}{\tau}\right) \tag{11}$$

The lifetime of $NO_x$ is variable and depends on ozone levels and the actinic flux. A typical lifetime of $NO_x$ is 3.8 h as shown in Liu et al. (2016), where lifetimes of urban $NO_x$ were estimated from satellite data. For simplicity, the lifetime of $NO_x$, $\tau$, is set to 3.8 h.

Taking into account these correction factors, the $NO_x$ emission rate $\boldsymbol{F_{NO_x}}$ is calculated by:

$$\boldsymbol{F_{NO_x}} = c_L \cdot c_\tau \oint_S VCD_{NO_2}(\boldsymbol{s}) \cdot \bar{\boldsymbol{w}} \cdot \bar{\boldsymbol{n}} \cdot \boldsymbol{ds} \tag{12}$$

To apply this method to our measurements, we have used the gridded data as shown in Figures 15 and 16. The measurement locations were converted to a Cartesian coordinate system, setting the origin in the center of Bucharest (Lat=43.4355°, Lon=26.1025°). Circles around this origin were defined with angular displacements in steps of 0.1°, resulting in 3600 sampling points. This step was performed for many radii in steps of 100 m, and an interpolated value at the circle locations was obtained. The $VCD_{NO_2}$ value for locations outside of the measured area, was set to the background value of $1 \times 10^{15}$ molec cm$^{-2}$.

Because of the discrete data, the integral is approximated by the sum of the sampled values.

$$\boldsymbol{F_{NO_x}} = c_L \cdot c_\tau \cdot \sum_i VCD_{NO_2}(s_i) \cdot \bar{w} \cdot \cos(\beta_i) \cdot \Delta s_i \tag{13}$$





Here, $\beta$ is the angle between the normal of the wind direction and the azimuth of the line segment and $\Delta s$ is the euclidean distance between the sample locations. The wind speed, $\bar{w}$ measured at Baneasa airport was used and set constant for each flight. The wind direction was determined from the apparent distribution of the plume. Table 8 lists the parameters used to analyze the two flights.

**Table 8.** Parameters used to calculate the $NO_x$ emission rate

| Parameter | 2014-09-08 | 2014-09-09 |
|---|---|---|
| Center coordinate | 44.4355°N, 26.1025°E | 44.4355°N, 26.1025°E |
| Wind speed ($\bar{w}$) | $1.1\,\mathrm{ms}^{-1}$ | $1.4\,\mathrm{ms}^{-1}$ |
| Wind direction | 57° | 65° |
| $c_L$ | 1.32 | 1.32 |
| $NO_x$ lifetime ($\tau$) | 3.8 h | 3.8 h |

Figure 21 shows the $NO_x$ emission rate of Bucharest determined from the method described above in dependence of distance to the city center. On both of the analyzed days, the determined emission rate increases until a distance of 10.9 km. At larger distances, the emission rate reaches a plateau. This behavior is related to the area covered by the flights, because at larger distances no measurements are available South of the center coordinate and the values are set to background values. Assuming background values is reasonable for the areas upwind (North), but is not appropriate for downwind areas (South) outside the measured domain, where enhanced $VCD_{NO_2}$ are expected, cf. Fig 15 and Fig 16. Thus, only emission rates up to a distance of 10.9 km give meaningful results. When considering the emissions within the radius of 10.9 km, the $NO_x$ emission rate is $15.1\,\mathrm{mol\,s}^{-1}$ on 2014-09-08 and $13.6\,\mathrm{mol\,s}^{-1}$ on 2014-09-09. The increase of the emission rate with distance may have two possible reasons: a) $NO_x$ has not yet reached its steady state ratio and b) the area where $NO_x$ is emitted increases, resulting in a larger emission rate. The latter is certainly true for this urban area, because the emissions do not only occur in the city center but across the whole extent of Bucharest. The effect mentioned first may explain the smaller slope at small distances to the origin.

Assuming an uncertainty of 33% on the $VCD_{NO_2}$, see 8.4 and an uncertainty on the wind speed of 50%, the overall uncertainty on the determined $NO_2$ emission rate is 60%. Shaiganfar et al. (2011) estimated the uncertainty of the correction factors, $c_L$ and $c_\tau$ to be 10% each. Applying these values also here, leads to a total uncertainty on the $NO_x$ emission rate of 62%.





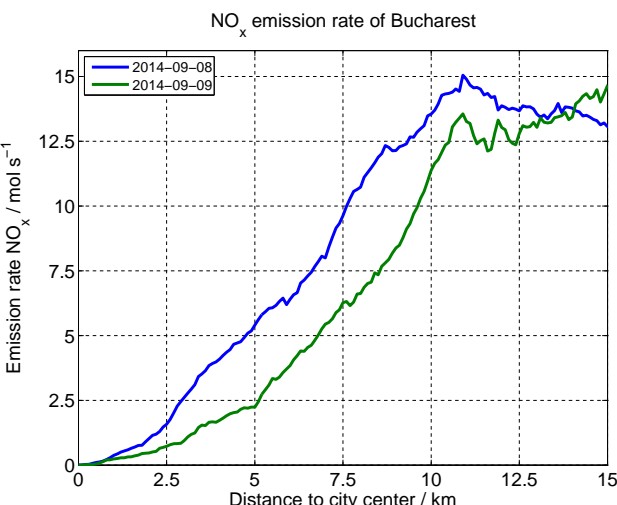

**Figure 21.** emission rate of $NO_x$ determined on the two investigated days. The center coordinate (source) was set to the city center.





## 11   Conclusions

In this paper, we presented airborne imaging DOAS measurements performed during the AROMAT campaign in September 2014. Two flights above Bucharest were performed, covering an area of about $18 \times 33 \, \mathrm{km}^2$ within $1.5 \, \mathrm{h}$ with a spatial resolution better than $100 \, \mathrm{m}$. These flights aimed at providing a high quality and fine resolution map of the horizontal $NO_2$ distribution above this large Eastern European city. To correct for the strongly varying surface reflectance within the city and its impact on the measurements, we have developed a method to derive surface reflectance information from an instrument which is not radiometrically calibrated. For this, we have used a look-up-table approach, in which an effective surface reflectance value is derived for each individual measurement from the measured relative intensity. A combination of MODIS retrievals over a reference region and SCIATRAN model data of atmospheric radiation was used to link relative intensities to absolute surface reflectances. The resulting scene specific surface reflectances have the advantage, that they directly match the measurements, avoiding artifacts from spatial sampling and interpolation and at least partially overcoming the need for precise knowledge of the surface BRDF. Comparison of measurements on the two days as well as observations taken under different relative azimuth angles shows excellent consistency of the derived surface reflectances. Further validation is planned by direct comparison to APEX measurements taken during a tandem flight over Berlin in April 2016. Using the AirMAP derived reflectance values, vertical columns were computed from the differential slant columns. While the $NO_2$ dSCD distribution shows spatial patterns related to surface properties, these are no longer observed in the $NO_2$ VCD distribution, indicating a successful correction for light path effects. Uncertainties in the AMF calculation were discussed in detail and inaccuracies in surface reflectivity and aerosol assumptions identified as main error sources. The latter could in principle be improved by aerosol soundings over the city center which were not possible during the AROMAT campaign. Strong spatial gradients in the $NO_2$ distribution could be observed in the covered area across the city, with $NO_2$ columns ranging from background values in rural areas upwind to about $4.2 \times 10^{16} \, \mathrm{molec \, cm^{-2}}$ over pollution hot spots. Measurements on subsequent days revealed quite distinct pollution patterns, probably related to changing meteorological conditions. Validation of the AirMAP observations with two independent co-located car-DOAS measurements performed on one of the measurement days shows good agreement between the datasets, indicating the good quality of the measurements. Using the AirMAP derived $NO_2$ distribution and wind data, the total $NO_x$ emission rate of Bucharest could be estimated to be about $14.4 \pm 8.9 \, \mathrm{mol \, s^{-1}}$. The airborne imaging DOAS measurements reported here illustrate the inhomogeneous and rapidly varying horizontal distribution of pollution on a city-scale that cannot be accessed by any other observation method at present. The measurements also illustrate the large sub-pixel variability in $NO_2$ data from present UV-vis satellite instruments like OMI and that AirMAP observations can be used for detailed validation of measurements from upcoming missions such as Sentinel 5 precursor having improved spatial resolution.



# Appendix A: Grid points of look-up-tables used RTM calculations

**Table 9.** Grid points in the AMF look-up table

| Parameter | Unit | Grid points |
| --- | --- | --- |
| Flight altitude (H) | km | 0.2, 0.3, 0.4, 0.5, 0.6, 0.7, 0.8, 0.9, 1, 1.1, 1.2, 1.3, 1.4, 1.5, 1.6, 1.7, 1.8, 1.9, 2, 2.1, 2.2, 2.3, 2.4, 2.5, 2.6, 2.7, 2.8, 2.9, 3, 3.1, 3.2, 3.3, 3.4, 3.5 |
| Ground surface reflectance (A) | | 0, 0.01, 0.02, 0.03, 0.04, 0.05, 0.06, 0.07, 0.08, 0.09, 0.10, 0.11, 0.12, 0.13, 0.14, 0.15, 0.16, 0.17, 0.18, 0.19, 0.20 |
| Viewing zenith angle (VZA) | ° | 0, 10, 20, 30, 40 |
| Relative azimuth angle (RAA) | ° | 0, 45, 90, 135, 180 |
| Solar zenith angle (SZA) | ° | 0, 10, 20, 30, 40, 50, 60, 65, 70, 75, 80 |
| Wavelength | nm | 310, 320, 330, 340, 350, 360, 370, 380, 390, 400, 410, 420, 430, 440, 450, 460, 470, 480, 490, 500 |

**Table 10.** Grid points in the look-up table of modeled intensities

| Parameter | Unit | Grid points |
| --- | --- | --- |
| Flight altitude (H) | km | 0.2, 0.3, 0.4, 0.5, 0.6, 0.7, 0.8, 0.9, 1, 1.1, 1.2, 1.3, 1.4, 1.5, 1.6, 1.7, 1.8, 1.9, 2, 2.1, 2.2, 2.3, 2.4, 2.5, 2.6, 2.7, 2.8, 2.9, 3, 3.1, 3.2, 3.3, 3.4, 3.5 |
| Ground surface reflectance (A) | | 0, 0.01, 0.02, 0.03, 0.04, 0.05, 0.06, 0.07, 0.08, 0.09, 0.1, 0.11, 0.12, 0.13, 0.14, 0.15, 0.16, 0.17, 0.18, 0.19, 0.2, 0.21, 0.22, 0.23, 0.24, 0.25, 0.26, 0.27, 0.28, 0.29, 0.3, 0.4, 0.5, 0.6, 0.7, 0.8, 0.9, 1 |
| Viewing zenith angle (VZA) | ° | 0, 10, 20, 30, 40 |
| Relative azimuth angle (RAA) | ° | 0, 10, 20, 30, 40, 50, 60, 70, 80, 90, 100, 110, 120, 130, 140, 150, 160, 170, 180 |
| Solar zenith angle (SZA) | ° | 0, 10, 20, 30, 40, 50, 60, 65, 70, 75, 80, 85, 90 |
| Wavelength | nm | 310, 320, 330, 340, 350, 360, 370, 380, 390, 400, 410, 420, 430, 440, 450, 460, 470, 480, 490, 500 |





*Acknowledgements.* The AROMAT campaign was funded by the European space agency, ESA. The authors would therefore like to thank ESA for enabling us to acquire this comprehensive dataset. Furthermore we thank Dirk Schüttemeyer for the successful coordination of the campaign. We also thank the Romanian authorities for their cooperation and the approval of the flights above Bucharest. Meteorological data was kindly provided by Meteo Romania. We thank the pilot of the aircraft, Carsten Lindemann, for his calm and professional flights, as well as his guidance in all matters related to the aircraft and the weather conditions. We acknowledge the work of the team around the MODIS satellite and the consortium in the ADAM project for providing the gridded surface reflectance data. Basemap data was extracted from ©OpenSteetMap contributors.





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
