# Peer review of "High-resolution airborne imaging DOAS-measurements of $NO_2$ above Bucharest during AROMAT"

_Atmospheric Measurement Techniques, 2016_

## Referee Comment (RC1) · Anonymous Referee #1 · 15 Dec 2016

The paper describes NO$_2$ measurements performed over Bucharest during the ARO-MAT campaign in September 2014. Nitrogen dioxide was observed with the Airborne imaging DOAS instrument AirMAP from the University Bremen. The focus of the paper is on the data retrieval and a short interpretation of the observed tropospheric column densities. To large parts it is interesting to read especially the albedo correction is very a nice idea. In some parts however the paper might be shortened a bit to stress out the main focus (AMF calculations with the highly varying albedo and the interpretation of the data). Therefore I suggest:

- Shifting the stratospheric correction (4.1.2) to the appendix. After half a page the authors state that the corrections discussed above were neglected here. From

[Figure]

Table 1 one can easily calculate that the difference in the stratospheric AMFs is close 0.2 for flight 2 and around 0.05 for flight 1. Moreover parts of this section focus on the satellite retrieval and are not directly related to the airborne instrument, though used for the correction in general.

- Shortening the description of the aerosol profile chosen and the other AMF parameters. I am not sure whether it suitable to use an aerosol profile from August 2015 for measurements in September 2014. But if a typical profile is assumed it might be used.

For the albedo correction two reference pixels from the MOD09A1 database, which has a resolution of 0.1 degree, are chosen. Why is the reference taken over the city? According to the map the intensity varies by a factor of 2 with high small scale variations. An area with less variability might be a better choice?

Airborne measurements provide high resolution and good data that might be used for many interesting scientific questions. The comparison to satellites is from my point of view not really an interesting question; we did that too often in the past. So I thank the authors that they did resisted this temptation.

**1  Detailed comments:**

- P3 l24 & p4 l1: The area of the city in km$^2$ and the area covered by the flights are not necessary. The maps show that most of the city including some suburbs were covered.

- P4 l1: Flight 1 (2014-09-08) started in the north and ended in the south, on this day the wind blew from north east (figure 2). Is it possible that the aircrafts plume was observed? The influence might be small and can hardly be estimated or?

- P4 l2: include "around local noon" in the flight description.

- P5 l18: The grating with 600 g/mm and 500 nm blaze wavelength was used. The other gratings (the Acton 300i allows up to 3) are not important in this context. The same grating was already used in Schönhardt et al. (2014).

- P5 l25: Does the size of the ground pixel change towards the edges of the swath? Later on it is mentioned for large roll angles of the plane but under normal flight conditions?

- P6 l3: There is an updated version of the solar atlas by Chance and Kurucz (2010), you might use instead.

- P7 l4: "multi measurements in one grid cell were averaged" how good do the multi measurements agree? At least for the vertical column that might be interesting to see.

- P10 l5: Please replace "normal to the surface" by "vertical", the local surface might differ from the horizontal e.g. mountains.

- P10 l13: The sentence beginning with "The dSCD and the AMF..." is not clear, please clarify.

- P13 figures 4 and 5: The BoxAMF for 440 nm are shown here and in section 5 the wavelength of 437.5 nm is mentioned (P17 l6), please use consistent data. Even though the difference in the AMF due to this wavelength change might be small,

- Figure 5: Is it useful to add a vertical line at 0.04 and 0.1 for a better comparison to Figure 4?

- P14 Figure 6: Is the figure necessary? In this figure it looks if the VZA is mainly in flight direction.

- P18 l6 Add a point "g)" For each measurement the retrieved albedo was used for the AMF calculation.

- P19 Figure 11 and P20 Figure 12: these two figures both show a linear correlation and a histogram. It would be nice if the same scales for the two histograms were used.

- P21 Figure 13: The grey and green areas are slightly confusing, on the other hand they might be important. Does it look better or worse if you show the street lines only?

- P21 l1-5: For each measurement the individual retrieved albedo was used. This might be mentioned a bit earlier in the analysis description e.g. P18 l6 g) (above)

- P25 figure 17: The measured wind directions does not agree with the apparent distribution of the plume, in the emission estimates the later wind direction is considered. Specify in the discussion of the figure.

- P25 l10 ff: The airborne measurements are very interesting by themselves. There is no need to compare to satellite data.

- P29 Table 6: just for completeness add the background correction here.

- P30 Table 7: Why is the uncertainty in the AMF calculation comparable, while MPIC uses a simple geometric approximation and the telescopes' elevation is $22°$ and the UGAL instruments points to the zenith and the AMF calculation is much more sophisticated. What is the typical resolution of these measurements for the local traffic and speed?

- P31 l10: The mismatch caused by the forward direction of the MPIC instrument's telescope was not corrected for?

- P33 l15 ff: The presented airborne measurements provide a much higher resolution than the OMI data used by Liu et al. (2016), moreover there are some "isolated" plumes like point source 9 in figure 15. If not for now but for the future it might be worth to estimate the NOx lifetime form the airborne measurements. Even the big Bucharest plume might be used, however here the assumption of a point source is no longer valid.

- P33 l20 ff: This circular approach seems appropriate for a point source. However a city is rather an area source than a point source (p34 l13). Also emissions upwind of the centre are to be considered. In the current approach these emissions are counted at a distance of l and it is not important if the distance is upwind or downwind of the city. But in a Cartesian system with one axis along the wind direction, integrating perpendicular to the wind, this effect can be taken into account. As the authors already mentioned, in the circular approach with radii steps of 100 m the area increases between 0.06 km$^2$ in the centre to 12.5 km$^2$ between 9.9 and 10 km (p34 l13).

**2  Typos and small corrections:**

- P2 l3: "processes" instead of "precesses"

- P27 l2 and l5: the surface reflectance for Figure 4 is given with 5% or 0.04, one of the numbers is not correct. In the figure label it is 0.04.

- P33 l21: Is it really a Cartesian coordinate system. In a first approximation the geographical system on that small scale is Cartesian. To me the new one looks like a polar system.

**3 References:**

For some references the doi is not given, it is not mandatory but good practise. Constantin 2012, Heue (2005 and 2008), Lohberger 2004 and Rozanov 2014. please recheck the others as well

For the two books Burrows, Borrell and Platt (2011) and Platt and Stutz (2008) there are brackets around the doi and doi is written in capital letters while in all other references it is not.

Chance, K., and Kurucz, R. L., An improved high resolution solar reference spectrum for earth's atmosphere measurements in the ultraviolet, visible, and near infrared. J. Quant. Spect. Rad. Trans., 111, 1289–1295, doi: 10.1016/j.jqsrt.2010.01.036, 2010.

---

## Referee Comment (RC2) · Anonymous Referee #2 · 16 Dec 2016

The authors describe measurements of NO2 above Bucharest by aircraft with an imaging UV/vis spectrometer. The paper focusses on the description of the retrieval chain and error assessment. The manuscript is suitable for publication in AMT after some minor corrections:

Major points:

Section 5: It is not fully clear to me from the text whether only the measured intensity of the centre wavelength of the NO2 fit window was used or the whole spectral range. I assumed so since it is explicitly mentioned for the model intensities. However, the measurements are supposedly a vector. But it is not mentioned what the dimension is. In the first read I assumed it was the wavelength. Please add some more explanations

in this section. Also, P. 16, L. 7: LUT of what? P. 19, L. 4: this is the first time that the text refers to the atmospheric correction explicitly.

Section 4.1.2: It's not explicitly mentioned that stratospheric NO2 varies with the SZA due to photochemical reactions. For this the time step of the model is important. Please also add to Eq. (4) that the VCDs depend on the time. What happens when a model grid cell border falls within the study area?

Technical and other small corrections:

P. 1, L. 9: I suggest rephrasing the part with 'at the aircraft'.

P. 1, L. 10: apparent instead of seen

P. 1, L. 16: Please state coincidence criteria and sample size here

P. 2, L. 26-30: The most important property here is the trace gas species and not the location of the measurements. I suggest amending these sentences accordingly.

P. 2, L. 35: Last sentence. Maybe add a sentence why that is better.

P. 3, L. 21: Formatting problems for reference

P. 3, L. 25: '. . .overlap of the swaths between adjacent flight tracks. . .' That's difficult to understand without the instrument description first.

P. 3, L. 26-28: Please add the local time of flight since this is important in relation to the SZA and rush hour.

P. 4, L. 3: DOAS was introduced before

P. 5, L. 10: 'For data safety reasons. . .' Is that due to memory effects of the CCD chip?

P. 5, Table 2: The FOV_along needs an aircraft speed to be meaningful.

P. 6, L. 1-7: I feel this information should be provided earlier on in the description. Also, I would sort the 4 sentences in the order of 2,1,3,4. Also, do you know what happens

at the edges of the pixels?

P. 6, L. 7: Remove 'even' and 'individual'.

P. 6, L. 11: how many adjacent rows?

P. 6, L. 18: 'as is the case for 'some' satellite measurements'

P. 6, L. 21: This back-scattered radiance spectrum. . .

P. 6, L. 23: Replace 'close' with 'slightly smaller or equal'. What are the expected background values in percent in comparison to the enhancements when polluted?

P. 6, L. 32: '. . ., whose pixel center.' Please rephrase

P. 7, Table 3: 'Fitted absorption cross-sections and 'other' important settings. . .'

P. 7, L. 2: 'in aircraft turns'

P. 7, L. 5: add comma after second date.

P. 7, L. 7: How many spectra do you add up for the background spectrum?

P. 7, L. 8: 'Some slightly negative values. . .'

P. 7, L. 9: That sentence implies that those high values are caused by the pollution which is not the case. Maybe rephrase?

P. 7, L. 11: with high values enhanced by up to 50% in comparison to the rest of the plume?

P. 10, L. 3: 'can be computed'

P. 10, L. 25: Maybe add a sentence after the equation that you are discussing the individual terms in the next section.

P. 11, L. 5: 'each individual measurement': Not really mentioned before that this is done and how many spectra are combined. See also comment above.

P. 11, L. 5-6: Only for the respective observation geometry? What about aerosols and albedo here?

P. 11, L. 6: Refer back to equation (2) here.

P. 11, L. 16: Please add reference to support this statement.

P. 11, L. 24: Formatting issue for reference

P. 12, Table 4: 500m layer for NO2: Is that a valid guess? Where does this number come from?

P. 12, L. 8: I would suggest moving this sentence 4 sentences down. Figure 4 only supports the statement of the first half of the previous sentence. Hence it is a bit misleading here.

P. 12, L. 8: BAMF: either explain or give reference here.

P. 14, Figure 6: I suggest updating this figure if the authors feel artsy: The way it is sketched now implies that the instrument on the plane is scanning across track.

P. 14, L. 2: RAA is already desribed on the page before.

P. 14, L. 9: 500m: see comment above.

P. 15, L. 4-5: Why are there no measurements available close to the ground?

P. 15, L. 6: Why was the profile with the lowest AOD chosen?

P. 16, L. 15: Remove 'unfortunately'.

P. 16, L. 16: Remove 2nd comma.

P. 16, L. 29-30: But the surface elevation surely was not constant?

P. 17, figure 8: What is cts?

P. 25, L. 13: Remove 'very'.

P. 26, L. 22: And what is the error on the ADAM reference surface reflectance?

P. 28, L. 3: remove comma after larger.

P. 28, L. 6, 8: 'antagonistic' is poor choice. Maybe 'opposing'.

P. 28, L. 5: Remove 2nd 'in a'.

P. 28, L. 10: Remove 'a'.

P. 29, L. 10: do you mention somewhere what the absolute stratospheric column is? Maybe repeat it here.

P. 31, L. 1: 'scatter is smaller'.

P. 31, L. 9: Does that time lag correspond to the speed of the car?

P. 32, figure 20: It's a bit misleading in this case here that the top panel in Figure (b) doesn't have markers on it since there is this large measurements gap. Maybe break the lines to show this. Why not make it panels A-C?

P. 33, L. 21: Polar coordinate system?

P. 36, L. 14: What is APEX?

---

## Author Comment (AC1) · 22 Mar 2017

**Author reply Referee #1**

We would like to thank referee #1 for his useful comments and detailed questions.
* * *
Legend:

- **Referee comment (bold)**

*Author's answer (italics)*

Modified manuscript text (blue)
* * *
The paper describes NO2 measurements performed over Bucharest during the AROMAT campaign in September 2014. Nitrogen dioxide was observed with the Airborne imaging DOAS instrument AirMAP from the University Bremen. The focus of the paper is on the data retrieval and a short interpretation of the observed tropospheric column densities. To large parts it is interesting to read especially the albedo correction is very a nice idea. In some parts however the paper might be shortened a bit to stress out the main focus (AMF calculations with the highly varying albedo and the interpretation of the data). Therefore I suggest:

- **Shifting the stratospheric correction (4.1.2) to the appendix. After half a page the authors state that the corrections discussed above were neglected here. From Table 1 one can easily calculate that the difference in the stratospheric AMFs is close 0.2 for flight 2 and around 0.05 for flight 1. Moreover parts of this section focus on the satellite retrieval and are not directly related to the airborne instrument, though used for the correction in general.**

*The stratospheric correction was not neglected in the analysis. The respective sentence reads: "As the measurements shown in this study were performed around noon, the diurnal variation in $SCD^{strat}$ are very small and could thus be neglected." We intended to use the subjunctive and did not state that we neglected the stratospheric correction. We acknowledge the ambiguity of that sentence and now changed it, see below. Since we have used the stratospheric correction in our retrieval, we prefer to keep this section in the main part of the manuscript.*

As the measurements shown in this study were performed around noon, the diurnal variations in $SCD^{strat}$ are very small and stratospheric correction is of minor importance.

*Considering the comment of Referee #2 we have also added the following two sentences to the text.*

For the two flights on 2014-09-08 and 2014-09-09 the stratospheric VCD was estimated to be around $3 \times 10^{15}$ molec cm$^{-2}$. The maximum change in the stratospheric SCD with respect to the reference spectrum, $\Delta SCD^{strat}$, was $1.5 \times 10^{14}$ molec cm$^{-2}$ and $3 \times 10^{14}$ molec cm$^{-2}$, respectively.

- **Shortening the description of the aerosol profile chosen and the other AMF parameters. I am not sure whether it suitable to use an aerosol profile from August 2015 for measurements in September 2014. But if a typical profile is assumed it might be used.**

*In order to improve the readability of that section we have now put the information about the profiles of aerosols (and NO$_2$) into separate subsections.*

*To answer the comment about the suitability of the used aerosol profile: Climatological data of the AOD in Bucharest is available on the AERONET website (https://aeronet.gsfc.nasa.gov/new_web/V2/climo_new/Bucharest_Inoe_500.html#2014). For September 2014, the monthly average of the AOD at 450nm is 0.26 +-0.1 (mean+-std). The used profile 31a (AOD=0.22) is the closest match of the experimental profiles to the climatological AOD value and is within the standard deviation of the monthly average. We thus assume that the used profile can be regarded as a typical profile for this location and season although the real aerosol profile during the measurements can of course be different. We have now added this information in the text.*

For the analysis of both flights in Sect. 7, we have used the profile FUBISS 31a. The AOD of this profile is the closest match to the monthly average AOD, available from AERONET measurements, having a value of AOD$_{450}$ = 0.26±0.1, (mean±stdev) (https://aeronet.gsfc.nasa.gov/new_web/V2/climo_new/Bucharest_Inoe_500.html#2014).

- **For the albedo correction two reference pixels from the MOD09A1 database, which has a resolution of 0.1 degree, are chosen. Why is the reference taken over the city? According to the map the intensity varies by a factor of 2 with high small scale variations. An area with less variability might be a better choice?**

*In general we would agree that an area with less variability would be a better choice. However, practical limitations make it almost impossible to find such an area.*

*1. The scaling factor ($I_{modRef} / I_{measRef}$) is determined for each viewing direction separately because the individual fibres have a non-uniform transmission. In order to discriminate the influence of outliers, such as high intensities caused by sun-glint or direct reflection it is desirable to average over many measurements. To determine the surface reflectance, the whole swath (>3km) must thus cover a homogeneous surface over a long distance along-track.*
*Possible candidates for surfaces with a homogeneous surface reflectance are (a) the lake in the north-west and (b) the forest in the north-east of the covered area. In the first attempts of the method development we tried to use these areas together with reflectance data from the USGS spectral library (https://speclab.cr.usgs.gov/spectral-lib.html) and the ASTER spectral library*

(http://speclib.jpl.nasa.gov/). However, using these areas and spectral libraries did not provide satisfying results.

The lake is not ideal because the water surface shows strong directional effects (cf. Fig 13 in paper) and thus the reflectance depends on the relative azimuth of the measurements. Furthermore its area is relatively small and does not allow averaging over many measurements.

The forest proved to be problematic for several reasons:

▪ The spectral databases contain spectra for different vegetation types, but the forest as a whole appears darker than individual leaves. This can be explained by the fact, that the tree canopy and the ground below absorb more radiation than an individual leave would. Using the reflectance values from the spectral libraries as $A_{Ref}$ thus resulted in too large surface reflectances.

▪ The vegetation in the forest area is not uniform. The woodland is interrupted by several forest glades (cf. figure below) having a different reflectance than the woodlands. When these forest glades are excluded from the surface reflectance reference area it cannot be assured that enough intensity measurements can be averaged.

[Figure]

2. We have chosen to use a global climatology to allow the application of the method in a consistent way for measurements in different regions and different times of the year. By using 0.1°x0.1° grid cells in the urban area a lot of AirMAP measurements (approx. 1300-2000 measurements per viewing direction) are averaged. This averaging makes the method more robust against outliers and directional effects of the surface. We have shown that the presented method gives consistent and satisfying results.

Another possibility would be to use satellite data on a daily basis. Reflectance data is available in relatively high spatial resolution from instruments such as MODIS (500m) or Landsat. However these

*datasets may also be useless if the overpasses on the days of the research flights are affected by clouds. Thus we have chosen to use a climatology, which is based on MODIS data.*

- **Airborne measurements provide high resolution and good data that might be used for many interesting scientific questions. The comparison to satellites is from my point of view not really an interesting question; we did that too often in the past. So I thank the authors that they did resisted this temptation.**

 **Detailed comments:**

•       **P3 l24 & p4 l1: The area of the city in km2 and the area covered by the flights are not necessary. The maps show that most of the city including some suburbs were covered.**

*Although this information may not be necessary for the interpretation of the data shown here, this information about the area covered and duration may be interesting for the reader. Thus we prefer to keep this information in the text.*

•       **P4 l1: Flight 1 (2014-09-08) started in the north and ended in the south, on this day the wind blew from north east (figure 2). Is it possible that the aircrafts plume was observed? The influence might be small and can hardly be estimated or?**

*If the aircraft's plume was observed during the flight, the influence must be small. Otherwise these emissions should show up in the dSCD results, especially in the clean regions. Either the $NO_2$ signal is too small to be detected by AirMAP, or the plume was not within AirMAP's field of view (FOV).*

*In order to check if the aircraft's plume could have been transported into AirMAP's FOV, we can estimate if the sum of the half swath width(w) (in upwind direction, at aircraft plume altitude) and the travelled distance by an air parcel into direction of the next flight line ($d_{South}$) is larger than the separation of the flight tracks ($d_{Track}$)*

$$test = (d_{South} + w) - d_{Track}$$

*If $test > 0$, then the aircraft's plume from the previous track was in AirMAP's FOV.*

*If $test \leq 0$, then the aircraft's plume from the previous track was not in AirMAP's FOV.*

*To calculate the displacement of exhaust $NO_x$ in southern direction the following formula can be used:*

$$d_{South} = time \times windSpeed \times \cos(windDir)$$

*On 2014-08-08, the wind was coming from 57°N with a speed of 1.1 $ms^{-1}$. The duration of one flight leg is about 10 minutes. Thus, the maximal time between adjacent measurement locations is 20 minutes. Plugging these values into the equation above yields:*

$$d_{South} = 1200s \times 1.1\ ms^{-1} \times \cos(57°)$$
$$= 719\ m$$

*The half swath width of AirMAP can be calculated as follows:*

$$w = H \times \tan\left(\frac{FOV_{across}}{2}\right)$$
$$= H \times \tan\left(\frac{51.7°}{2}\right)$$
$$= H \times 0.48$$

, where $H$ is the vertical distance to the object of interest (the possible plume). Because the aircraft altitude was constant, the vertical distance to the plume is determined by the vertical movement of the air, i.e. the distance an air parcel has travelled into ground direction. Because the speed of the downward vertical transport is not known, it is simply set to the same value as the horizontal wind speed. Using this assumption, an air parcel may be transported into ground direction with:

$$d_{vertical} = 1200s \times 1.1\ ms^{-1} = 1320m$$

Setting $H$ to $d_{vertical}$ results in a half swath width of $w = 634\ m..$

The distance between adjacent tracks is about $d_{Track} = 1900m$.

Now these values can be plugged into the first equation to see whether the aircraft's plume could be observed.

$$test = (719m + 634m) - 1900m$$
$$= 1353m - 1900m$$
$$= -547m$$

This result is much smaller than 1 indicating that an observation of the aircraft's plume is rather unlikely because the locations of the outermost viewing direction and the exhaust $NO_x$ are separated by more than 500m.

The approach developed above has of course many uncertainties involved. First of all, the horizontal wind speed at flight altitude is probably larger than the measured wind speed on ground. On the other hand, the vertical wind speed is usually much lower than the horizontal wind speed. However, even if the air masses could have been transported into AirMAP's field of view, we do not see any influence on our measurements.

- **P4 l2: include "around local noon" in the flight description.**

Both flights were performed around local noon under cloud free and sunny conditions with low wind speeds (< 1.5 ms$^{-1}$ )

- **P5 l18: The grating with 600 g/mm and 500 nm blaze wavelength was used. The other gratings (the Acton 300i allows up to 3) are not important in this context. The same grating was already used in Schönhardt et al. (2014).**

The 300g/mm grating was also used during the campaign, but not for the flights presented here. In future publications we may show results, in which the 300g/mm grating was used for measurements in the UV spectral range. In order to inform the reader about the availability of that different instrumental setup we prefer to keep that information in the text.

- **P5 l25: Does the size of the ground pixel change towards the edges of the swath? Later on it is mentioned for large roll angles of the plane but under normal flight conditions?**

*Image distortions, such as barrel or cushion distortion were not characterized in the lab. However, the manufacturer's specification states that the objective has a low distortion. Thus it is assumed that the instantaneous field of view of the viewing directions is constant in terms of solid angle. When the footprint is projected on the ground it becomes larger with increasing VZA. This increase affects the across-track width, as well as the along-track width of the ground footprint. The across-track width of a ground pixel can be described by the difference of the tangens between the VZA of the outer vertex (VZA$_{outer}$) and the VZA of the inner vertex (VZA$_{inner}$) of a viewing direction i.*

$$PixelWidth_{across} = H \times \tan(VZA_{i_{outer}}) - \tan(VZA_{i_{inner}})$$

The instantaneous pixel width in along-track direction can be described by:

$$iPixelWidth_{along} = \frac{1}{\cos(VZA_i)} \times H \times 2 \times \tan\left(\frac{iFOV_{along}}{2}\right)$$

Where H is the altitude a.g.l. and iFOV$_{along}$ is the instantaneous field-of-view in along-track direction.

*The figure below shows the increase in pixel width with increasing VZA normalized to the central viewing direction and illustrates that the across-track pixel width increases by about 20% for the outer viewing directions during a level flight (not in turns). The along track pixel width increases by about 10%.*

[Figure]

- **P6 l3: There is an updated version of the solar atlas by Chance and Kurucz (2010), you might use instead.**

*Thank you for the suggestion. We will test the updated version of the solar atlas. However, we only use the solar atlas, convoluted with the slit function for the calibration of the spectra. Considering the spectral resolution of the instrument, the changes are expected to be very small.*

- **P7 l4: "multi measurements in one grid cell were averaged" how good do the multi measurements agree? At least for the vertical column that might be interesting to see.**

*Variability between multiple measurements in one grid cell can have several reasons. The measurements are affected by fitting noise, directional effects of the surface, differences in light path from geometry (sun and instrument), imperfect geolocation and of course temporal variability in $NO_2$.*

*A possibility to investigate how good the averaged measurements agree is to look at the standard deviation of the measurements per grid cell. A map of the standard deviation within the grid cells is shown in Figure 1 (below) for the results from the flight on 2014-09-08. Note that the range of the color scale was reduced by a factor of 10 as compared to the $NO_2$ VCD map in the paper. Figure 2 (below) shows the number of measurements that were averaged in each grid cell. Grid cells that only contain measurements of one flight line generally average 2 measurements (along track). Grid cells in the regions of overlapping swaths from adjacent flight tracks contain about twice as many measurements. The standard deviation of the $NO_2$ VCD in the regions of non-overlapping swaths (i.e. where N=2 or N=3) is about 1.7 x $10^{15}$ molec/cm$^2$. In areas of overlapping swaths (N>=4), the standard deviation is 2.0 x $10^{15}$ molec/cm$^2$. The value of the standard deviation of the whole dataset is 1.6 x $10^{15}$ molec/cm$^2$. In the case of grid cells corresponding to one flight line, these values can be explained to a large extent by the fit error of the DOAS analysis. When taking the mean of the quotient of the dSCD fit error of $NO_2$ and the corresponding AMF the result is:*

$$\frac{1}{n} \times \sum \frac{fiterror_{NO_2}}{AMF} = 1.44 \times 10^{15} molec\ cm^{-2}$$

*This value is a bit smaller than the value of the standard deviation for the non-overlapping swath regions. However, some natural variability of the $NO_2$ field also contributes to the variability of one grid cell. The standard deviation in the regions of overlapping swaths is larger, which is probably caused by natural variability of $NO_2$ between successive overpasses and by the fact that the ground spot is observed "from the other side" (from North to South, then South to North). This means that the air mass over which the measurement integrates is not the same.*

*Computing standard deviations for grid cells with only 2 to 4 samples may not be a good measure. An alternative approach is to compute absolute differences between multiple overpasses. For this we have separated the measurements according to the heading of the aircraft. Two data grids were produced,. One with $NO_2$ VCDs when the aircraft was heading to the East (yaw=90°+-5°), and one with $NO_2$VCDs when the aircraft was heading to the West (yaw=270°+-5°). This approach does not consider variability of measurements in one grid cell that were averaged along track, but the standard deviation can be computed from a much larger sample size. A histogram of absolute differences (East-West) is shown in*

Figure 3*(below). From almost 12000 compared grid cells, we find a mean close to zero and a standard deviation of 2.9 x 10$^{15}$ molec cm$^{-2}$.*

[Figure]

**Figure 1: Standard deviation (σ) of the NO2 VCD average per grid cell. Data is taken from the flight on 2014-09-08.**

[Figure]

**Figure 2: Number of measurement averaged per grid cell. Data is taken from the flight on 2014-09-08.**

[Figure]

**Figure 3: Histogram of absolute differences between co-located NO$_2$ VCD from different overpasses during the same flight.**

- **P10 l5: Please replace "normal to the surface" by "vertical", the local surface might differ from the horizontal e.g. mountains.**

The change in light path length, as compared to a vertical path, is usually expressed in the form of an air mass factor (AMF), which is defined as the ratio of slant and vertical column densities:

- **P10 l13: The sentence beginning with "The dSCD and the AMF. . ." is not clear, please clarify.**

In these equations the terms dSCD and AMF refer to the trace gas amount fitted from a single spectrum and its corresponding air mass factor.

- **P13 figures 4 and 5: The BoxAMF for 440 nm are shown here and in section 5 the wavelength of 437.5 nm is mentioned (P17 l6), please use consistent data. Even though the difference in the AMF due to this wavelength change might be small,**

*We have computed new BAMF at the center wavelength of the fit window (437.5 nm). The new computations also used a finer altitude grid. Figures 4 & 5 were updated accordingly.*

- **Figure 5: Is it useful to add a vertical line at 0.04 and 0.1 for a better comparison to Figure 4?**

*Figure 5 was updated to show the AMF at 437.5 nm. Additionally the suggested vertical lines at 0.04 and 0.1 were added to support the reader.*

- **P14 Figure 6: Is the figure necessary? In this figure it looks if the VZA is mainly in flight direction.**
*We have updated the figure.*

- **• P18 l6 Add a point "g)" For each measurement the retrieved albedo was used for the AMF calculation.**
*Thank you for this suggestion. We have added the following sentence after point f).*
The above procedure yields scene specific surface reflectances for each measurement which are later used in the AMF calculations.

- **P19 Figure 11 and P20 Figure 12: these two figures both show a linear correlation and a histogram. It would be nice if the same scales for the two histograms were used.**
*Thank you for the suggestion. The histograms of Fig. 11 & Fig 12 now have the same scale.*

- **P21 Figure 13: The grey and green areas are slightly confusing, on the other hand they might be important. Does it look better or worse if you show the street lines only?**
*We have now plotted the map on a gray background to make the map more clear and less overloaded. Figure 13 was updated accordingly.*

- **P21 l1-5: For each measurement the individual retrieved albedo was used. This might be mentioned a bit earlier in the analysis description e.g. P18 l6 g) (above)**
*Done. See answer for comment P18 l6.*

- **P25 figure 17: The measured wind directions does not agree with the apparent distribution of the plume, in the emission estimates the later wind direction is considered. Specify in the discussion of the figure.**
*Thank you for this remark. We have now added the following sentence in the discussion of the figure:*
Comparing the apparent wind direction, as seen in the $NO_2$ distribution, to the data record of the meteorological station at Baneasa airport it is worth to note that there is good agreement on 2014-09-09, whereas a mismatch of the wind direction is observed on 2014-09-08.

- **P25 l10 ff: The airborne measurements are very interesting by themselves. There is no need to compare to satellite data.**
*We are happy to hear that.*

- **P29 Table 6: just for completeness add the background correction here.**
*Table 6 was updated to include uncertainties related to the correction of the tropospheric background.*

•	**P30 Table 7: Why is the uncertainty in the AMF calculation comparable, while MPIC uses a simple geometric approximation and the telescopes' elevation is 22 and the UGAL instruments points to the zenith and the AMF calculation is much more sophisticated. What is the typical resolution of these measurements for the local traffic and speed?**

a.	*Uncertainties: The uncertainties stated were taken from the respective publications (Shaiganfar et al. 2011) for the MPIC retrieval and (Constantin et al. 2013) for the UGAL retrieval.*

b.	*Resolution: The spatial resolution of the car measurements depends on the integration time used in the data acquisition and the speed of the car. Because the measurements were performed in an urban area, the speed is variable.*

*The statistics of the spatio-temporal resolution of the car measurements are summarized in the table below. The figure below the table shows the data from which these valued were computed.*

*In case of the MPIC instrument, which points to an elevation of 22°, the spatial resolution is coarser than the distance driven between consecutive measurements due to horizontal averaging. The magnitude of the spatial averaging depends on the height of the $NO_2$ profile. As already stated in Eq. 9 (manuscript), the horizontal averaging can be described by:*

$$dist_{ave} = \frac{Height_{NO_2}}{\tan(elevationAngle)}$$

*Assuming a box profile of 500 m, this results in a horizontal averaging of about 1.2 km, which has to be added to the distances driven by the MPIC car.*

| Parameter | MPIC | UGAL |
|---|---|---|
| **Temporal Resolution (mean) [s]** | 77 | 32 |
| **Temporal Resolution (median) [s]** | 62 | 32 |
| **Temporal Resolution (std) [s]** | 33 | 3 |
| **Spatial resolution (mean) [m]** | 423 | 285 |
| **Spatial resolution (median) [m]** | 348 | 276 |
| **Spatial resolution (std) [m]** | 447 | 218 |

[Figure]

**Figure 4: Upper panel: Temporal resolution of the car measurements (left: MPIC, right: UGAL). Lower panel: Spatial resolution i.e distance traveled by the car between the measurements**

- **P31 l10: The mismatch caused by the forward direction of the MPIC instrument's telescope was not corrected for?**

*No, in the current approach the car's position was used for both instruments as co-location criterion. The comparison between airborne measurements and off-axis car-DOAS measurements may be improved in future analyses, taking into account the azimuth and the elevation angle of the measurements as well as the $NO_2$ profile.*

- **P33 l15 ff: The presented airborne measurements provide a much higher resolution than the OMI data used by Liu et al. (2016), moreover there are some "isolated" plumes like point source 9 in figure 15. If not for now but for the future it might be worth to estimate the NOx lifetime form the airborne measurements. Even the big Bucharest plume might be used, however here the assumption of a point source is no longer valid.**

*That would be an interesting research topic, indeed. However, for this paper such an analysis is out of scope.*

- **P33 l20 ff: This circular approach seems appropriate for a point source. However a city is rather an area source than a point source (p34 l13). Also emissions upwind of the centre are to be considered. In the current approach these emissions are counted at a distance of l and it is not important if the distance is upwind or downwind of the city. But in a Cartesian system with one axis along the wind direction, integrating perpendicular to the wind, this effect can be taken into account. As the authors already mentioned, in the circular approach with radii steps of 100 m the area increases between 0.06 km2 in the centre to 12.5 km2 between 9.9 and 10 km (p34 l13).**

*In our current approach emission upwind of the city center are considered as negative contribution to the total emission rate. We do not integrate the $NO_2$ over the area inside the circle, but along the circumference of the circle (line), see also P32 l13. The circle was only used to define the sampling points at different distances from the center. In that sense the method is similar to car-DOAS measurements when driving around the city on a circular road. The inflow of $NO_2$ rich air is accounted for in the term $cos(\beta)$, which is the angle between the normal of the wind direction and the azimuth of the line segment (P34 L1). This ensures that $NO_2$ transported into the encircled area becomes negative and has a negative contribution to the outflow. This is illustrated by the following figure which shows the term $cos(\beta)$ (color coded), the wind direction and the normal to the wind direction for the flight on 2014-09-08 at an example radius of 5km.*

The inflow of NO2 enriched air is accounted for in the term $cos(\beta)$, which is the angle between the normal of the wind direction and the azimuth of the line segment. This term ensures that NO2 transported into the encircled area becomes negative and does not contribute to the emissions determined from within the circle. The term $\Delta s$ is the Euclidean distance between the sample locations.

[Figure]

**2 Typos and small corrections:**

•	**P2 l3: "processes" instead of "precesses"**
*Fixed.*

•	**P27 l2 and l5: the surface reflectance for Figure 4 is given with 5% or 0.04, one of the numbers is not correct. In the figure label it is 0.04.**
*Corrected.*

•	**P33 l21: Is it really a Cartesian coordinate system. In a first approximation the geographical system on that small scale is Cartesian. To me the new one looks like a polar system.**
*We have used a Cartesian coordinate system. First, the grid of NO$_2$ VCDs was converted to Cartesian coordinates. Then the sampling angles $\phi$, were defined from 0° to 360° in steps of 0.1°. In the next step, the sampling locations were obtained by conversion from polar to Cartesian coordinates, using the respective radius R.*

$$x = \sin\left(\phi \cdot \frac{\pi}{180}\right) \cdot R$$
$$y = \cos\left(\phi \cdot \frac{\pi}{180}\right) \cdot R$$

**3 References:**

•	**For some references the doi is not given, it is not mandatory but good practice.**
•	**Constantin 2012**

*Unfortunately this publication does not have a DOI.*

•	**, Heue (2005 and 2008),**
•	**Lohberger 2004**
•	**and Rozanov 2014.**
*Fixed.*

**please recheck the others as well**

•	**For the two books Burrows, Borrell and Platt (2011) and Platt and Stutz (2008) there are brackets around the doi and doi is written in capital letters while in all other references it is not.**
•	**Chance, K., and Kurucz, R. L., An improved high resolution solar reference spectrum for earth's atmosphere measurements in the ultraviolet, visible, and near infrared. J. Quant. Spect. Rad. Trans., 111, 1289–1295, doi: 10.1016/j.jqsrt.2010.01.036, 2010.**

---

## Author Comment (AC2) · 22 Mar 2017

**Author reply Referee #2**

We would like to thank referee #2 for his useful comments and detailed questions.
* * *
Legend:

- **Referee comment (bold)**

*Author's answer (italics)*

Modified manuscript text (blue)
* * *
**The authors describe measurements of NO2 above Bucharest by aircraft with an imaging UV/vis spectrometer. The paper focusses on the description of the retrieval chain and error assessment. The manuscript is suitable for publication in AMT after some minor corrections:**

**Major points:**

- **Section 5: It is not fully clear to me from the text whether only the measured intensity of the centre wavelength of the NO2 fit window was used or the whole spectral range. I assumed so since it is explicitly mentioned for the model intensities. However, the measurements are supposedly a vector. But it is not mentioned what the dimension is. In the first read I assumed it was the wavelength. Please add some more explanations in this section. Also, P. 16, L. 7: LUT of what? P. 19, L. 4: this is the first time that the text refers to the atmospheric correction explicitly.**

*The recorded spectra are measurement vectors containing the measured intensity (unit: Counts) as a function of wavelength. These spectra are analyzed using the DOAS technique to retrieve trace gas slant column densities. In the surface reflectance retrieval we do not use the full resolution spectra, but the average intensity in the fitting window, normalized to an illumination time of 1 second. The quantity 'intensity' used for the surface reflectance retrieval is thus a scalar (for each spectrum), with the unit Counts $s^{-1}$. In the table-lookup, the center wavelength of the fit window (plus all geometry variables) is used to obtain the modeled intensity from the LUT of modeled intensities.*

The measured spectra contain information on the recorded light intensity during the exposure. The scalar quantity 'intensity' is computed for each spectrum, which is the average light intensity in the fitting window of the DOAS analysis, normalized to an illumination time of one second.

[….]

The influence of the observation geometry and the surface reflectance on the intensities measured by AirMAP were modeled for a given aerosol scenario (green line in Fig. 7) with the SCIATRAN RTM and compiled into a look-up-table (LUT) of modeled intensities at the aircraft. This LUT of modeled intensities is used to correct for atmospheric effects affecting the intensities measured by AirMAP. In order to isolate the contribution of the surface reflectance to the measured intensities, a 6 step procedure (a-f) is applied.

- **Section 4.1.2: It's not explicitly mentioned that stratospheric NO2 varies with the SZA due to photochemical reactions. For this the time step of the model is important. Please also add to Eq. (4) that the VCDs depend on the time. What happens when a model grid cell border falls within the study area?**

  *In order to obtain B3dCTM model values for an individual measurement, linear interpolation is used in all three dimensions (Latitude,Longitude, Time). The variation of the stratospheric $NO_2$ is characterized by an almost linear increase of the $NO_2$ VCD in the sunlit atmosphere. The temporal resolution of the B3dCTM model is 10 minutes. Due to the high temporal resolution, and the linear behavior of the stratospheric $NO_2$ we think that the temporal evolution of the $NO_2$ VCD is captured well. By using the linear interpolation in latitude and longitude, no abrupt changes are expected when a grid cell border intersects the study area.*

  *We have changed the text to include the information on the temporal resolution.*

The B3dCTM model provides stratospheric VCDs of $NO_2$ on a global grid in a spatial resolution of 2.5°x3.75° and a temporal resolution of 10 minutes.

*We have added the time variable to Eq.4. The use of linear interpolation in space and time is now explicitly mentioned in the text.*

A model value is obtained for each location (lat,lon) and time (t) of the measurements using linear interpolation in space and time.

**Technical and other small corrections:**

- **P. 1, L. 9: I suggest rephrasing the part with 'at the aircraft'.**
  As the instrument is not radiometrically calibrated, we have developed a method to derive the surface reflectance from intensities measured by AirMAP.

- **P. 1, L. 10: apparent instead of seen**
  While surface properties are clearly apparent in the NO 2 dSCD results, this effect is successfully corrected for in the VCD results.

- **P. 1, L. 16: Please state coincidence criteria and sample size here.**
  *We understand that information about coincidence criteria and sample size is needed to interpret the comparison to the car measurements. Providing this information in the abstract may however be too much text for this section. This information is given later in the text (Section 9). Thus we prefer to keep that sentence unchanged.*

- **P. 2, L. 26-30: The most important property here is the trace gas species and not the location of the measurements. I suggest amending these sentences accordingly.**
  *That is a good comment in order to support the reader. These sentences now read:*
  In more recent years, imaging DOAS (iDOAS) instruments were developed. Lohberger et al. (2004) demonstrated the applicability for trace gas retrievals in a ground-based setup. Installed on aircrafts, these systems enable the creation of maps of the horizontal trace gas distribution. iDOAS measurements of anthropogenic point source emissions of $NO_2$ were performed by Heue et al. (2008) and by Schönhardt et al. (2015) above the South African Highveld plateau and above a German power plant, respectively. . General et al. (2014) performed measurements of volcanic emissions of BrO, OClO, and $SO_2$ at Mt. Etna, Italy. Urban NO 2 distributions were measured by Popp et al. (2012) above the city of Zürich, Switzerland, and by Lawrence et al. (2015) above the city of Leicester, England.

- **P. 2, L. 35: Last sentence. Maybe add a sentence why that is better.**
  Mapping of $NO_2$ distributions above cities with airborne iDOAS provides a holistic picture on pollution levels across the city and may be used identify the contribution of different $NO_x$ sources, such as industry and traffic, to pollution levels in a city.

- **P. 3, L. 21: Formatting problems for reference**
  *Fixed.*

- **P. 3, L. 25: '. . .overlap of the swaths between adjacent flight tracks. . .' That's difficult to understand without the instrument description first.**
  *In this part of the text we describe the research flights and the flight patterns. In this section, the reader shall be informed that the flights aimed at producing gapless maps. Further explanations about AirMAP's viewing geometry follow in Section 3.1. A reference to Sect 3.1 is now added to the text.*
  This pattern provides sufficient overlap of the swaths between adjacent flight tracks to produce a gap-less map (see also Sect. 3.1).

- **P. 3, L. 26-28: Please add the local time of flight since this is important in relation to the SZA and rush hour.**
  *The times of the flight are listed in Table 1. The footer of Table 1 the states that: "Local time (EEST) is UTC+3".*
  *The sentence describing the condition was now changed to include the information, that the flights were performed 'around local noon'.*
  Both flights were performed around local noon under cloud free and sunny conditions with low wind speeds

- **P. 4, L. 3: DOAS was introduced before.**
  *Fixed.*

The Airborne imaging Differential Optical Absorption Spectroscopy instrument for Measurements of Atmospheric Pollution (AirMAP) has been developed for the purpose of trace gas measurements and pollution mapping.

- **P. 5, L. 10: 'For data safety reasons. . .' Is that due to memory effects of the CCD chip?**
  *No, this procedure is not related to a memory effect. The data on the CCD is stored in a circular buffer. In order to prevent possible data loss, for example from buffer overflow, the circular buffer is flushed every minute (60.7s). In this way, no more than one minute of data is lost by such a problem. This is what was meant by "data safety reasons".*

- **P. 5, Table 2: The FOV_along needs an aircraft speed to be meaningful.**
  *Table 2 now states the instantaneous field of view of the instrument. The instantaneous FOV is independent of the aircraft speed.*

- **P. 6, L. 1-7: I feel this information should be provided earlier on in the description. Also, I would sort the 4 sentences in the order of 2,1,3,4. Also, do you know what happens at the edges of the pixels?**
  *In this section we first describe the instrumental setup and later the viewing geometry. It is a matter of personal taste which topic should be discussed first. The subsequent section describes the post-processing of the CCD images to individual viewing directions. Therefore we think that it is better to keep the two parts (viewing geometry & post-processing) close together.*

  *With increasing VZA the pixels become larger. This increase of the pixel width in across-track dimension can be up to 20% for the outermost viewing direction during a level flight. See also answers to Referee #1.*

- **P. 6, L. 7: Remove 'even' and 'individual'.**
  With the AirMAP setup it is thus possible to examine the sub-pixel variability within one OMI pixel (13×24km$^2$ at nadir), or a S5p-satellite pixel (3.5×7km$^2$ at nadir).

- **P. 6, L. 11: how many adjacent rows?  → (P. 5 L. 31?)**
  *The number of averaged adjacent rows is on average (SizeCCD/#ViewingDirection=512/35 = 14.6). Due to small variations in the cross-section of the individual light fibres and the cladding in the fibre bundle, the exact number of averaged rows depends on the extent of an individual light fibre imaged on the CCD and ranges from 13 to 15 rows.*

- **P. 6, L. 18: 'as is the case for 'some' satellite measurements' → P.6 L.7**
  Using an extraterrestrial solar spectrum as background spectrum in the DOAS analysis, as is done in some satellite retrievals, yields slant column densities (SCDs), which are the number densities of an absorber, integrated along the light path.

- **P. 6, L. 21: This back-scattered radiance spectrum.      → P.6 L.8**
  This back-scattered radiance spectrum may contain small amounts of the absorber.

- **P. 6, L. 23: Replace 'close' with 'slightly smaller or equal'. What are the expected background values in percent in comparison to the enhancements when polluted? → P.6 L.10**
  *In our retrieval we assume a background $NO_2$ VCD of $1x10^{15}$ molec/cm$^2$. Considering a moderate pollution level (of the presented flights) of $2x10^{16}$ molec/cm$^2$, the $NO_2$ contained in the background spectrum has a share of 5% to the signal in polluted areas. Stating that the dSCD is equal to the SCD, thus does not seem appropriate. 'Slightly smaller' however may be better than 'close'. The sentence now reads:*
  Because the background spectrum is taken at a clean location, the dSCD is only slightly smaller than the SCD.

- **P. 6, L. 32: '. . ., whose pixel center..' Please rephrase → P7.L3**
  The measurements are spatially binned by the pixel centers. All measurements with pixel-center coordinates falling into a grid cell are assigned to that grid cell. Multiple measurements in one grid cell are averaged using the unweighted arithmetic mean.

- **P. 7, Table 3: 'Fitted absorption cross-sections and 'other' important settings. . .'**
  Fitted absorption cross-sections and important other settings used in the retrieval of $NO_2$ dSCDs.

- **P. 7, L. 2: 'in aircraft turns'       → P7. L6**
  This effect can be observed in aircraft turns when the projected footprint becomes larger.

- **P. 7, L. 5: add comma after second date.**
  *Fixed.*

- **P. 7, L. 7: How many spectra do you add up for the background spectrum?**
  *Table 2 lists that the exposure time is 0.5s. In table Table 3 we write that the background spectrum is a 60s average (also P 6 L. 7).. Consequently 120 individual spectra are averaged for the background spectrum. This information was now included in the text in Sect. 3.2.*
  The background spectrum $I_0$ used in the DOAS analysis of the AirMAP spectra is an average over 60 seconds  (120 individual spectra) taken from a scene of the same flight having low absorber abundances.

- **P. 7, L. 8: 'Some slightly negative values. . .'**
  Some slightly negative values occur in the background region as a result of instrumental noise, because the dSCD-values are scattered around zero.

- **P. 7, L. 9: That sentence implies that those high values are caused by the pollution which is not the case. Maybe rephrase?**
  *The maximal dSCD value is caused by high pollution levels in conjunction with bright surfaces, which enhance the $NO_2$ signal. The dSCDs are impacted by many factors, which are addressed in the derivation of the VCDs (Sect. 4).*

*A new sentence was added, which emphasizes the influence of bright surface on the measured dSCDs. See answer to next comment.*

- **P. 7, L. 11: with high values enhanced by up to 50% in comparison to the rest of the plume?**
  *A detailed examination of the derived surface reflectance in the region, highlighted in Figure 2, reveals surface reflectances of around 0.03 for non-built-up areas and surface reflectances in the range of of 0.08-0.14 for the 'bright' pixels. Taking the data from Figure 5, an increase in surface reflectance from 0.03 (AMF=1.4) to 0.1(AMF=2.1) results in an AMF increased by 50%. This indicates that enhanced values of 50% in comparison to the rest of the plume can be explained by the surface reflectance.*
  *We have now added more explanations to the text.*
  *[…]*
  Some lightly negative values occur in the background region as a result of instrumental noise, because the dSCD-values are scattered around zero. In the most polluted areas, dSCD-values of up to $6.1 \times 10^{16}$ moleccm$^{-2}$ are observed. The dSCDs show a plume of $NO_2$ spreading south-westwards from the city center. The $NO_2$ field inside the plume shows small-scale structures with high values. Some of these structures, e.g. the pronounced values at the ring road in the South-West, are not associated with $NO_2$ emissions from traffic, but are related to bright surfaces. These bright surfaces can enhance the $NO_2$ dSCDs by about 50% compared to neighboring pixels, having a darker surface, see also Fig. 5 in Sect. 4.2.
  *[…]*

- **P. 10, L. 3: 'can be computed'**
  Using the dSCDs resulting from the DOAS fits, vertical column densities (VCDs), defined as the absorber concentration integrated along the vertical direction, can be computed.

- **P. 10, L. 25: Maybe add a sentence after the equation that you are discussing the individual terms in the next section.**
  *We have now added the sentence:*
  The individual terms introduced in Eq. 2 and Eq. 3 are discussed in the following subsections.

- **P. 11, L. 5: 'each individual measurement': Not really mentioned before that this is done and how many spectra are combined. See also comment above**
  *The calculation of the AMF for the background spectrum is performed in the same way as for all other measurements. This means that all parameters listed in Table 4 of the paper are accounted for. Besides the flight altitudes and the angles, this includes the aerosol and $NO_2$-profile as well as the scene-specific surface reflectance. The text now includes more information.*

  *[…]*
  The background spectrum is an averaged spectrum from 120 individual spectra, which may have different AMFs caused by changing conditions (geometry, surface reflectance) between these measurements. For each individual measurement during the integration time of the background spectrum, the AMF is computed for the respective condition during a single exposure, as will be

shown in Sect. 4.2. The AMF of each single spectrum is multiplied with the $VCD_0^{trop}$, see Eq. 2. The average of the product is used as the tropospheric part of the reference background ($SCD_0^{trop}$).

- **P. 11, L. 5-6: Only for the respective observation geometry? What about aerosols and albedo here?**
  *See previous answer.*

- **P. 11, L. 6: Refer back to equation (2) here.**
  *Done.*

- **P. 11, L. 16: Please add reference to support this statement.**
  *The bias between SCIAMACHY measurements the stratospheric model, OSLO CTM2, was investigated in* (Hilboll et al., 2013)*. The method presented there was applied to investigate the performance of the B3dCTM model. Results can be found in* (Hilboll, 2014)*.*
  *The text now includes these references.*

- **P. 11, L. 24: Formatting issue for reference**
  *Fixed.*

- **P. 12, Table 4: 500m layer for NO2: Is that a valid guess? Where does this number come from?**
  *We agree that the justification of the assumed $NO_2$ profile was too short. We have now provided more explanations to justify the assumed $NO_2$ profile.*

  No information about the $NO_2$ vertical distribution is available for the conditions of the flights. Thus assumptions have to be made. In order to keep the definition of the unknown profile simple, we have chosen to use a box profile with a homogenous mixing ratio. The altitude in which the $NO_2$ resides depends on many parameters, such as emission altitude, boundary layer height, orography, temperature etc. Vlemmix et al., 2015 derived $NO_2$ profiles from MAX-DOAS measurements in China, showing that the $NO_2$ profile height is between 500 m and 1000 m. Mendolia et al., 2013 studied the urban $NO_2$ profile of Toronto and found the average characteristic profile height to be around 500 m during summer. These studies suggest that the assumption of a 500m $NO_2$ layer is a reasonable guess.

- **P. 12, L. 8: I would suggest moving this sentence 4 sentences down. Figure 4 only supports the statement of the first half of the previous sentence. Hence it is a bit misleading here.**
  *We prefer to keep that sentence where it is.*

- **P. 12, L. 8: BAMF: either explain or give reference here.**
  *Added citation.*
  For a detailed explanation of the BAMF concept see Rozanov and Rozanov (2010) and references therein.

- **P. 14, Figure 6: I suggest updating this figure if the authors feel artsy: The way it is sketched now implies that the instrument on the plane is scanning across track.**
  *We have updated the figure.*

- **P. 14, L. 2: RAA is already described on the page before.**
  *Thank you for this comment. The RAA was not the only angle explained a second time… The changed text segment now reads:*
  The observation geometry relevant for the RTM calculations is described by three angles: Solar Zenith Angle (SZA), Viewing Zenith Angle (VZA) and Relative Azimuth Angle (RAA). Figure 6 illustrates the meaning of these angles. The VZA is the deviation from the direct nadir observation geometry. As the VZA increases, the light paths get longer. The VZA changes with the viewing direction, but is also altered with the aircraft's attitude. The RAA is the difference between the Solar Azimuth Angle (SAA) and VAA of the measurement.. Following the SCIATRAN convention, the RAA is defined as 0° if the instrument is pointed towards the sun (forward scattering) and 180° for the direction away from the sun (backward scattering). The SZA is the angle between the zenith and the center of the sun's disc and impacts on the length of the light path through the Earth's atmosphere.

- **P. 14, L. 9: 500m: see comment above.**
  *See answer above.*

- **P. 15, L. 4-5: Why are there no measurements available close to the ground?**
  *This is a general problem of the 2 types of measurements.*
  *The Raman-Lidar measurements do not provide measurements in the lowest few hundred meters. This is mainly caused by geometrical reasons, because the telescope cannot capture the entire backscattered signal from close distances.*
  *The FUBISS-ASA2 airborne sun-photometer has to point directly towards the sun. Vertical aerosol information can only be derived during vertical motion of the aircraft. If the instrument cannot be pointed towards the sun during ascent / descent, no information can be retrieved. The instrument is pointing always to the right with respect to the flight direction. The flight track during ascend and descend follows the regulations of the airspace and cannot be chosen by the instrument operator. Thus it is often not possible to point the instrument towards the sun, especially during take-off and landing.*

- **P. 15, L. 6: Why was the profile with the lowest AOD chosen?**
  *Climatological data of the AOD in Bucharest is available on the AERONET website (https://aeronet.gsfc.nasa.gov/new_web/V2/climo_new/Bucharest_Inoe_500.html#2014). For September 2014, the monthly average of the AOD at 450nm is 0.26 +-0.1 (mean+-std). The used profile 31a (AOD=0.22) is the closest match to the climatological value and is within the standard deviation of the monthly average. We thus assume that the used profile can be regarded as a typical profile for this location and season. We have now added this information in the text.*

For the analysis of both flights in Sect. 7, we have used the profile FUBISS 31a. The AOD of this profile is the closest match to the monthly average AOD, available from AERONET measurements, having a value of $AOD_{450}$ = 0.26±0.1, (mean±stdev) (https://aeronet.gsfc.nasa.gov/new_web/V2/climo_new/Bucharest_Inoe_500.html#2014).

- **P. 16, L. 15: Remove 'unfortunately'.**
  *Done.*

- **P. 16, L. 16: Remove 2nd comma.**
  *Done.*

- **P. 16, L. 29-30: But the surface elevation surely was not constant?**
  That is true. The surface elevation was not constant. However, the area in and around Bucharest has a flat topography. At a constant flight altitude, modulations in the altitude a.g.l are only caused by buildings. In general the buildings have a height of about 10m or 20m. These modulations are rather small in relation to the flight altitude of 3360m a.g.l.. Thus changes in the measured intensities, caused by different altitudes can be neglected here.

- **P. 17, figure 8: What is cts?**
  *'Cts' was an abbreviation for 'counts'. We have now changed the labels of Figs. 8, 11, 12  and write the full word without abbreviations.*

- **P. 25, L. 13: Remove 'very'.**
  *Done.*

- **P. 26, L. 22: And what is the error on the ADAM reference surface reflectance?**
  *The surface reflectance database also provides uncertainties associated with each grid cell, which is based on the variability of the MODIS measurements. For both of the surface reflectance reference grid cells a variance of $1.83 \times 10^{-5}$ is given. Thus, the uncertainty (1sigma) of the surface reflectance value is 0.0043. With a surface reflectance of 0.0394, this results in a relative uncertainty of 11%. The uncertainty is now stated in the text.*

  Two grid cells of the ADAM database were used as the reference region, both having a surface reflectance value $A_{Ref}$ of 0.0394±0.0043.

- **P. 28, L. 3: remove comma after larger.**
  *Done.*

- **P. 28, L. 6, 8: 'antagonistic' is poor choice. Maybe 'opposing'.**
  *Changed antagonistic to opposing*

- **P. 28, L. 5: Remove 2nd 'in a'.**
  *Done.*

- **P. 28, L. 10: Remove 'a'.**
  *Done*.

- **P. 29, L. 10: do you mention somewhere what the absolute stratospheric column is? Maybe repeat it here.**
  *We have now added information about the absolute stratospheric column, as well as information on the impact of the stratospheric correction in Sect. 4.1.2 (Stratospheric correction)*

  As the measurements shown in this study were performed around noon, the diurnal variations in $SCD^{strat}$ are very small and stratospheric correction is of minor importance.
  [....]
  For the two flights on 2014-09-08 and 2014-09-09 the stratospheric VCD was estimated to be around 3 x $10^{15}$ molec cm$^{-2}$. The maximum change in the stratospheric SCD with respect to the reference spectrum, $\Delta SCD^{strat}$, was 1.5 x $10^{14}$ molec cm$^{-2}$ and 3 x $10^{14}$ molec cm$^{-2}$, respectively.
  .

- **P. 31, L. 1: 'scatter is smaller'.**
  When comparing to the UGAL instrument the scatter is smaller, presumably because the spatial inhomogeneities in the $NO_2$ field affect the comparison to a much smaller extent because the instrument is pointed to the zenith.

- **P. 31, L. 9: Does that time lag correspond to the speed of the car?**
  *The speed of the car surely impacts on the 'time lag', but is not the only reason. The 'time lag' discussed is caused by errors in the geolocation of the compared $NO_2$ measurements. We have used the car's location as co-location criterion which is a point. However, the measurements were not only performed at that location but along the route of the car (line) during the integration time. If the $NO_2$ field is not homogeneous, this results in spatial misregistrations. Furthermore, the MPIC measurements were performed under an elevation angle of 22°. In the manuscript we have assessed the potential spatial mismatch between the car's position and the location of the measured $NO_2$ signal. The comparison between airborne measurements and off-axis car-DOAS measurements may be improved in future analyses, taking into account the azimuth as well and the elevation angle and the $NO_2$ profile. For such a correction of the geolocation, car-DOAS measurements with a higher temporal resolution would be desirable to avoid changes of the azimuth during the integration time.*

- **P. 32, figure 20: It's a bit misleading in this case here that the top panel in Figure (b) doesn't have markers on it since there is this large measurements gap. Maybe break the lines to show this. Why not make it panels A-C?**
  *We excluded the markers, because the plot became messy and overloaded. Breaking the lines is a good suggestion and we have changed the plot accordingly.*

- **P. 33, L. 21: Polar coordinate system?**

  *We have used a Cartesian coordinate system. First, the grid of $NO_2$ VCDs was converted to Cartesian coordinates. Then the sampling angles $\phi$, were defined from 0° to 360° in steps of 0.1°. In the next step, the sampling locations were obtained by conversion from polar to Cartesian coordinates, using the respective radius R.*

  $$x = \sin\left(\phi \cdot \frac{\Pi}{180}\right) \cdot R$$
  $$y = \cos\left(\phi \cdot \frac{\Pi}{180}\right) \cdot R$$

- **P. 36, L. 14: What is APEX?**

  *APEX is the "Advanced Prism EXperiment" and was already introduced on P.15 L.14 in the sentence:*

  *To our knowledge, no data product provides information on the ground spectral surface reflectance in sufficient spatial resolution to be used for our measurements. Such data is only acquired on campaign basis from instruments such as APEX (Itten et al., 2008) or HySpex (Baumgartner et al., 2012).*